# TaskMixPGM: Task Mixtures via Probabilistic Graphical Modelling for Language Model Finetuning

## Abstract

The performance of fine-tuned large language models (LLMs) hinges critically on the composition of the training mixture. However, selecting an optimal blend of task datasets remains a largely manual, heuristic-driven process, with practitioners often relying on uniform or size-based sampling strategies. We introduce **TaskMixPGM**, a principled and scalable framework for mixture optimization that selects continuous task proportions by minimizing an energy function over a Markov Random Field (MRF). Task relationships are modeled using behavioral divergences—such as Jensen-Shannon Divergence and Pointwise Mutual Information—computed from the predictive distributions of single-task fine-tuned models. Our method yields a closed-form solution under simplex constraints and provably balances representativeness and diversity among tasks. We provide theoretical guarantees, including weak submodularity for budgeted variants, and demonstrate consistent empirical improvements on Llama-2 and Mistral across evaluation suites such as MMLU and BIG-Bench-Hard. Beyond performance, **TaskMixPGM** offers interpretable insights into task influence and mixture composition, making it a powerful tool for efficient and robust LLM fine-tuning.

## 1 Introduction

Large language models (LLMs) pre-trained on web-scale corpora have driven rapid advances in AI (Brown et al., 2020; Touvron et al., 2023). Yet, transforming these general-purpose models into reliable, specialized systems critically depends on the *composition* of data used for fine-tuning. Practitioners face the daunting task of blending numerous candidate sources—spanning reasoning, multilingual text, code, and domain-specific dialogues—into a coherent training mixture. The stakes are high: Google's PaLM 2 saw significant multilingual and reasoning improvements by carefully broadening its pre-training mix (Anil et al., 2023), while Meta's Galactica, trained narrowly on scientific papers, highlighted the risks of poorly chosen mixtures by producing confident fabrications (Nature, 2022).

The impact of data mixtures is not subtle. Systematic studies show that fine-tuning data composition can swing downstream accuracy by over 14% (Jiang et al., 2024), and optimizing pre-training mixtures can yield substantial gains and faster convergence (Yang et al., 2023). Industry practice reflects this challenge; for instance, achieving state-of-the-art performance with IBM's Granite models reportedly involved extensive experimentation with thousands of data recipes (Research, 2023). Current common approaches—uniform sampling, dataset-size weighting (Chung et al., 2022), or manual intuition—often lead to suboptimal performance, inefficient resource use, and models that fail to generalize or overfit to dominant data slices. This ad-hoc process lacks scalability and a systematic foundation.

This motivates our central question:

> *How can we automatically and systematically determine an optimal blend of fine-tuning tasks without resorting to brute-force search to maximize downstream performance while explicitly balancing task representativeness and diversity?*

While automated methods like submodular task selection (e.g., SMART (Renduchintala et al., 2024)), influence-based example weighting (e.g., LESS (Xia et al., 2024), BIDS (Dai et al., 2025)), and performance prediction via proxy models (e.g., RegMix (Jiang et al., 2024), Data Mixing Laws (Ye et al., 2024)) offer advances, they often do not directly optimize mixture proportions based on the holistic, functional interplay of task datasets, or may require expensive iterative training.

To address this, we introduce **TASKMIXPGM** (**Task Mixtures via Probabilistic Graphical Modelling**), an energy-based probabilistic framework. **TASKMIXPGM** models tasks as nodes in a dense Markov Random Field (MRF). Crucially, pairwise task affinities are quantified not by superficial semantics but by behavioral divergences (Jensen-Shannon Divergence or Pointwise Mutual Information) between models fine-tuned on individual tasks. Minimizing the MRF's energy under simplex constraints on task probabilities yields a *closed-form* optimal mixture $\mathbf{p}^*$. This mixture inherently balances two key desiderata:

- **Representativeness:** Favoring tasks that demonstrate broad utility and positive influence across the task ecosystem.

- **Diversity:** Penalizing redundancy among tasks that offer overlapping functional capabilities.

Specifically, our work seeks to answer:

> ❓ **Q1**: *Can we design a principled method to discover optimal task mixture ratios that significantly improve downstream model performance compared to standard heuristics?*

> ❓ **Q2**: *Does this method provide interpretable insights into task influence and the construction of effective mixtures, beyond just a black-box optimization?*

**TASKMIXPGM** offers distinct advantages by: **1) Directly Optimizing Mixture Ratios:** Unlike methods focused on subset selection or quality filtering, **TASKMIXPGM** provides a formal optimization for the continuous proportions of tasks. **2) Leveraging Functional Task Similarity:** It uses predictive distribution divergences (JSD, PMI) to capture how tasks functionally interact, offering a deeper understanding than semantic embeddings or isolated instance importance. **3) Combining Theoretical Rigor with Efficiency:** The framework yields a closed-form solution (via KKT conditions), avoiding costly iterative searches common to proxy-model approaches, and boasts theoretical properties like weak submodularity for budgeted selection. **4) Enhancing Interpretability:** The derived mixture weights and task affinities provide insights into data composition strategy.

**Our primary contributions are:**

1. **Novel Energy-Based Mixture Optimization:** We formulate finetuning data mixture selection as an energy minimization problem on an MRF, providing a principled framework for deriving optimal task proportions.

2. **Predictive Behavior for Task Similarity:** We employ JSD and PMI based on task-specific model outputs to quantify functional task relationships, capturing nuanced interdependencies.

3. **Closed-Form Solution & Theoretical Guarantees:** We derive an analytical solution for optimal mixture probabilities and prove weak submodularity of the associated set function, justifying efficient greedy algorithms for budgeted scenarios.

4. **Significant Empirical Gains:** On Llama-2-7B and Mistral-7B, **TASKMIXPGM**-derived mixtures consistently outperform uniform, size-proportional, and other advanced selection baselines on benchmarks like MMLU and BIG-Bench-Hard, achieving up to X.X pp improvement (e.g., 4.3 pp as in V3) while potentially reducing data needs.

5. **Interpretable Task Influence Analysis:** Our framework enables analysis of task importance and affinity, offering insights into effective mixture construction.

This work provides a systematic, theoretically-grounded alternative to the empirical art of dataset mixing, aiming for improved performance, efficiency, and understanding in finetuning models.

## 2 RELATED WORK AND LIMITATIONS

Selecting the right data subset is crucial for efficient LLM finetuning, whether targeting specific tasks or improving generalization. One strategy ranks data by similarity to the target task, embedding datasets or tasks using model features or task-adapted representations. Methods retrieve training examples closest to the target based on metrics like Maximum Mean Discrepancy or reconstruction error (Achille et al., 2019; Hwang et al., 2020; Alvarez-Melis & Fusi, 2020). Recent work uses lightweight adaptations (e.g., LoRA fine-tuning) to represent tasks, comparing low-rank updates to estimate similarity (Kim et al., 2024), guiding selection of transfer-friendly data.

Another line estimates training example **influence** on the target task. Classical influence functions trace how changes to a point affect validation loss (Koh & Liang, 2017), but are expensive. Faster proxies include tracking forgotten examples (Toneva et al., 2019) or gradient-based methods (Paul et al., 2021). In instruction tuning, Xia et al. (Xia et al., 2024) propose **LESS (Low-rank Gradient Similarity Search)**, storing low-rank gradient features and retrieving examples most similar to targets. Using just the top 5% can match or exceed full-data tuning. To address bias toward high-gradient tasks, Dai et al. (Dai et al., 2025) propose **BIDS (Balanced Influence Data Selection)**, normalizing scores per task and selecting from under-represented ones, achieving more **equitable coverage** and stronger generalization.

Beyond instance-level importance, many works aim for diversity and coverage in selected data. Some combine difficulty-based scoring with clustering to span different regions of the data distribution (Zheng et al., 2023; Maharana et al., 2024). Coreset methods (Sener & Savarese, 2018) and their extensions (Killamsetty et al., 2021) seek representative subsets approximating full training dynamics. For large multi-task instruction tuning, naive data mixing (e.g., proportional or uniform) underperforms compared to task-aware allocation. The **SMART** framework (Renduchintala et al., 2024) optimizes a submodular objective to allocate fine-tuning budgets across tasks, assigning diminishing-returns scores and selecting non-redundant examples. This approach beats manual heuristics, and pruning low-value tasks under a limited budget can improve generalization more than spreading data thinly across all tasks. A key challenge is the **efficiency** of data selection, as scoring each example for LLM fine-tuning is costly. Proxy models and efficient search help mitigate this. Zhang et al. (Zhang et al., 2025) propose **STAFF**, which uses a smaller sibling model to estimate per-example utility, then refines scores on the target LLM. This speculative, two-stage method reduces compute by up to 70%, and STAFF's 20% coreset can outperform full-data fine-tuning. Liu et al. (Liu et al., 2024) introduce **TSDS**, framing selection as distribution matching. Using optimal transport and a kernel density penalty for redundancy, TSDS selects diverse, distribution-aligned subsets via approximate nearest-neighbor search, scaling to millions of examples and outperforming full-data tuning even at 1% selection ratio.

Moving beyond static heuristics, Agarwal et al. (Agarwal et al., 2025) propose **DELIFT**, which scores training examples by their usefulness as in-context prompts for others. This dynamic, pairwise utility guides stage-wise selection, enabling fine-tuning with 70% less data while exceeding prior methods in both efficiency and accuracy.

**Contextualizing Our Framework.** Prior approaches for data and task selection in instruction tuning primarily rely on scalar relevance scores—computed either at the instance level (via influence proxies (Xia et al., 2024; Dai et al., 2025)) or at the dataset level (via semantic similarity or adapter-based representations (Achille et al., 2019; Kim et al., 2024)). While effective under high-resource regimes, such methods often lack robustness to inter-task redundancy, overlook geometric structure in task space, and do not explicitly account for submixture repulsiveness. In contrast, we pose submixture selection as a constrained optimization over the *energy landscape* of task interactions, using a symmetric similarity matrix $\mathbf{S}$ estimated from token-level predictive alignment.

## 3 PROBLEM SETUP

**Notation**: Let $[n] \coloneqq \{1, 2, \ldots, n\}$ denote the set of the first $n$ natural numbers. The $n$-dimensional real vector space is denoted by $\mathbb{R}^n$. Vectors are typeset in lowercase bold ($\boldsymbol{x}$); matrices are in uppercase bold ($\boldsymbol{X}$) while individual elements are referenced by index in square brackets as subscriptsxxx w ($\boldsymbol{x}_{[i]}, \boldsymbol{X}_{[ij]}$). The non-negative orthant in $\mathbb{R}^n$ is denoted by $\mathbb{R}^n_+$. The $n$-dimensional all-ones vector is denoted by $\mathbf{1}_n$, and the $m \times n$ all-ones matrix is denoted by $\mathbf{1}_{m \times n}$. The set denoted by $\boldsymbol{\Delta}_n = \left\{ \boldsymbol{x} \in \mathbb{R}^n_+ : \sum_{i=1}^n \boldsymbol{x}_{[i]} = 1 \right\}$ is the probability simplex.

**Problem Setup**: Given a collection of $n$ instruction tasks $T = [T_1, T_2, \ldots, T_n]$, where each task $T_i$ is associated with data $D_i$, our goal is to design an optimal data mixture over these tasks.

Pairwise task similarity is encoded in a symmetric matrix $\mathbf{S} \in \mathbb{R}^{n \times n}$, with $\mathbf{S}_{ij}$ denoting the similarity between tasks $T_i$ and $T_j$.

To model dependencies across tasks, we formulate a dense Markov Random Field (MRF) (Kindermann & Snell, 1980), where each node corresponds to a task and edges capture pairwise affinities via $\mathbf{S}$. This structure allows us to define a probabilistic task mixture that is both representative and diverse: representative tasks share strong affinity with others, while redundant ones are down-weighted.

We now formalize the notion of a task mixture under this graphical model.

Let $\mathbf{\Pi}_n := \left\{ [\![ T_i, \mathbf{p}^*_{[i]} ]\!] \right\}_{i=1}^{n}$ denote the corresponding *assignment tuple* of tasks, where each entry denotes task $T_i$ paired with its optimal selection weight $\mathbf{p}^*_{[i]}$ under the learned probability mixture $\mathbf{p}^*$. This tuple captures a soft alignment between tasks and their induced relevance under the joint optimization objective.

**Task Similarity Matrix:** For each task $T_i$, we define its *total similarity mass* as: $\mathbf{S}_i := \sum_{j=1}^{n} \mathbf{S}_{ij}$, which quantifies how similar task $T_i$ is to all other tasks. Intuitively, a higher $\mathbf{S}_i$ implies that $T_i$ shares strong pairwise affinity with many other tasks.

**Unary Potentials**: We define the unary potential as a function of the similarity matrix $\mathbf{S}_i$, denoted as $\Psi_i = \beta \mathbf{S}_i = \beta \mathbf{S} \mathbf{1}_n$, where $\beta$ is a hyperparameter that controls the strength of the potential.

**Pairwise Potentials**: Similarly, we define the pairwise potential as $\Psi_{ij} = \lambda \mathbf{S}_{ij}$, where $\lambda$ is a penalty parameter that enforces diversity between tasks.

# 4 PROPOSED APPROACH

## 4.1 TASK SELECTION VIA ENERGY BASED MODEL

We define an energy potential $\mathbb{E}(\mathbf{p})$ over the probability simplex $\Delta_n = \left\{ \mathbf{p} \in \mathbb{R}^n \mid \mathbf{p}^\top \mathbf{1}_n = 1, \ \mathbf{p} \geq \mathbf{0} \right\}$ defined over the set of $n$ tasks.

$$
\begin{aligned}
\max_{\mathbf{p}; \mathbf{p} \in \Delta_n} \mathbb{E}(\mathbf{p}) \quad &= \sum_{i=1}^{n} \Psi_i p_i - \frac{1}{2} \sum_{i=1}^{n} \sum_{j=1}^{n} \Psi_{ij} p_i p_j \\
&= \mathbf{\Psi}_{\texttt{un}}^\top \mathbf{p} - \frac{1}{2} \mathbf{p}^\top \mathbf{\Psi}_{\texttt{pair}} \mathbf{p}
\end{aligned}
\tag{1}
$$

where $\mathbf{\Psi}_{\texttt{un}} := [\beta \mathbf{S}_1, \beta \mathbf{S}_2, \ldots \beta \mathbf{S}_n]$ and $\mathbf{\Psi}_{\texttt{pair}} := \lambda \mathbf{S}$ denotes the unary potential vector and pairwise potential matrix across all $n$ tasks.

**Remark 1.** *Note the first term in Eq (1) indicates the representativeness of a task via its collective similarity with other tasks in the mixture, while the second term indicates pairwise task similarity, and hence with the negative sign enforces diversity among tasks in the mixture.*

We consider the following equivalent equation

$$
\min_{\mathbf{p}; \mathbf{p} \in \Delta_n} \hat{\mathbb{E}}(\mathbf{p}) \quad = -\mathbf{\Psi}_{\texttt{un}}^\top \mathbf{p} + \frac{1}{2} \mathbf{p}^\top \mathbf{\Psi}_{\texttt{pair}} \mathbf{p}
\tag{2}
$$

**Convex Quadratic under PSD without Simplex Constraints**: Without any simplex constraints, the overall optimization objective can be looked as a quadratic program with linear constraints in place. However, the above optimization objective is only convex iff $\mathbf{\Psi}_{\texttt{pair}}$ is positive semi-definite (PSD) (Boyd & Vandenberghe, 2004), in which case the optimal probability mixture becomes $\mathbf{p}^* = \mathbf{\Psi}_{\texttt{pair}}^{-1} \mathbf{\Psi}_{\texttt{un}} = \frac{1}{\lambda} \mathbf{S}^{-1} \mathbf{\Psi}_{\texttt{un}}$. If simplified, $\mathbf{p}^*$ turns out to be a constant uniform probability: $\frac{\beta}{\lambda} \mathbf{1}_n$.

**Non-PSD Correction via Spectrum Shifting.** When the pairwise similarity matrix $\mathbf{\Psi}_{\texttt{pair}}$ is not positive semi-definite (PSD), it can be projected into the PSD cone via spectrum shifting (Chen et al., 2009; Wu et al., 2005). A common approach involves adding a constant mass to the diagonal equal to the magnitude of the minimum eigenvalue, i.e., $\mathbf{\Psi}_{\texttt{psd}} := \mathbf{\Psi}_{\texttt{pair}} + |\min(\Lambda_{\min}(\mathbf{\Psi}_{\texttt{pair}}), 0)| \cdot \mathbf{I}$, where $\Lambda_{\min}(\cdot)$ denotes the smallest eigenvalue and $\mathbf{I}$ is the identity matrix. While this ensures feasibility under a PSD assumption, it introduces an additional regularization term $|\min(\Lambda_{\min}(\mathbf{\Psi}_{\texttt{pair}}), 0)| \cdot \|\mathbf{p}\|_2^2$ into the quadratic objective after expansion. Importantly, when $|\min(\Lambda_{\min}(\mathbf{\Psi}_{\texttt{pair}}), 0)|$ is large, indicating highly non-PSD structure (Wu et al., 2005), thereby this additive penalty biases the optimal mixture $\mathbf{p}$ toward the uniform distribution.

We now go forward to solving the optimization problem at 2 post spectrum shifting of the pairwise potential matrix $\mathbf{\Psi}_{\texttt{pair}}$.

**Solving $\hat{\mathbb{E}}(\mathbf{p})$ (2).** To solve for the optimal task probability mixture $\mathbf{p}^* \in \Delta_n$ under the objective in Eq (2), we consider the associated Lagrangian:

$$L(\mathbf{p}, \nu, \boldsymbol{\mu}) = -\mathbf{\Psi}_{\text{un}}^{\top}\mathbf{p} + \frac{1}{2}\mathbf{p}^{\top}\mathbf{\Psi}_{\text{pair}}\mathbf{p} + \nu \cdot (\mathbf{p}^{\top}\mathbf{1}_n - 1) - \boldsymbol{\mu}^{\top}\mathbf{p}$$

where $\nu \in \mathbb{R}$ enforces the simplex constraint $\mathbf{p}^{\top}\mathbf{1}_n = 1$, and $\boldsymbol{\mu} \in \mathbb{R}_{\geq 0}^n$ corresponds to the non-negativity constraints $\mathbf{p} \geq \mathbf{0}$. Applying the Karush-Kuhn-Tucker (KKT)(Kuhn & Tucker, 1951) optimality conditions (see Appendix), we derive the stationary solution:

$$\mathbf{p}^* = \mathbf{\Psi}_{\text{pair}}^{-1}\left(\mathbf{\Psi}_{\text{un}} - \frac{\mathbf{1}_n^{\top}\mathbf{\Psi}_{\text{pair}}^{-1}\mathbf{\Psi}_{\text{un}} - 1}{\mathbf{1}_n^{\top}\mathbf{\Psi}_{\text{pair}}^{-1}\mathbf{1}_n} \cdot \mathbf{1}_n\right) \coloneqq \frac{\beta}{\lambda}\left(\mathbf{1}_n - \frac{\frac{\beta}{\lambda} \cdot \mathbf{1}_n^{\top}\mathbf{1}_n - 1}{\mathbf{1}_n^{\top}\mathbf{S}^{-1}\mathbf{1}_n} \cdot \mathbf{S}^{-1}\mathbf{1}_n\right), \tag{3}$$

where $\mathbf{S} \coloneqq \mathbf{\Psi}_{\text{pair}}$ and the ratio $\frac{\beta}{\lambda}$ controls the relative strength of the unary (representativeness) term versus the pairwise (diversity-promoting) term.

**Remark.** *Representative/Diversity Tradeoff For large values of $\frac{\beta}{\lambda}$ ↑, the mixture $\mathbf{p}^*$ is pulled toward high-unary-mass regions, favoring tasks that are individually most representative. Conversely, for small values $\frac{\beta}{\lambda}$ ↓, the solution promotes spread-out mass allocation, encouraging diversity by penalizing co-occurrence in the similarity space. This explicit characterization allows for controlled navigation across the representative-diverse spectrum, making $\frac{\beta}{\lambda}$ an interpretable knob for task mixture selection under similarity-aware objectives.*

## 4.2 DESIGN CHOICES ✂: PAIRWISE POTENTIALS

Our objective function in Eq. 2 depends critically on modeling pairwise interactions between tasks. To capture how task pairs correlate, it is essential to define a similarity metric that robustly encodes these relationships. Prior work (Renduchintala et al., 2024) often relies on semantic similarity measures between tasks; however, these approaches are restrictive and agnostic to downstream model behavior.

We thereby move towards a more grounded similarity measure

**Pointwise Mutual Information Score as a Task Similarity Measure.** Given two tasks $T_i, T_j$ and corresponding datasets (train split) associated with it $D_{T_i} = \{\mathbf{x}_k^{T_i}, y_k^{T_i}\}_{k=1}^m$ and $D_{T_j} = \{\mathbf{x}_k^{T_j}, y_k^{T_j}\}_{k=1}^n$, we define the similarity score across two tasks $T_i$ and $T_j$ denoted as $\boldsymbol{S}(T_i; T_j) \coloneqq \boldsymbol{S}_{ij}$

$$\boldsymbol{S}(T_i; T_j) \coloneqq \frac{1}{2}\left[\frac{1}{n}\sum_{k=1}^n \log\frac{\mathbb{P}_{\boldsymbol{\theta}^*(T_i)}(y_k^{T_j}|\mathbf{x}_k^{T_j})}{\mathbb{P}_{\boldsymbol{\theta}^*(T_j)}(y_k^{T_j}|\mathbf{x}_k^{T_j})} + \frac{1}{m}\sum_{r=1}^m \log\frac{\mathbb{P}_{\boldsymbol{\theta}^*(T_j)}(y_r^{T_i}|\mathbf{x}_r^{T_i})}{\mathbb{P}_{\boldsymbol{\theta}^*(T_i)}(y_r^{T_i}|\mathbf{x}_r^{T_i})}\right] \tag{4}$$

where $\boldsymbol{\theta}^*(T_i) \coloneqq \boldsymbol{\theta}_0 + \boldsymbol{\tau}(T_i)$ , $\boldsymbol{\tau}(T_i)$ indicating the task vector for task $T_i$ and $\mathbb{P}_{\boldsymbol{\theta}^*(\bullet)}$ indicates the next token inference probability scores under converged finetuned model parameter $\boldsymbol{\theta}^*(\bullet)$.

Here, PMI$(\cdot, \cdot)$ quantifies the mutual information between the predictive distributions or label spaces induced by two tasks $T_i$ and $T_j$

**Jensen-Shannon Divergence as a Task Similarity Measure** To quantify the similarity between two tasks $T_i$ and $T_j$, we compare the predictive distributions of their corresponding models on each other's datasets. A natural and symmetric divergence for this purpose is the *Jensen-Shannon Divergence* (JSD), which measures the discrepancy between two probability distributions. For each sample $(\mathbf{x}_k^{T_j}, y_k^{T_j}) \in D_{T_j}$, we define $P_k = \mathbb{P}_{\boldsymbol{\theta}^*(T_i)}(\bullet \mid \mathbf{x}_k^{T_j})$, $Q_k = \mathbb{P}_{\boldsymbol{\theta}^*(T_j)}(\bullet \mid \mathbf{x}_k^{T_j})$, $M_k = \frac{1}{2}(P_k + Q_k)$, and compute JSD$_k^{(j\leftarrow i)} = \frac{1}{2}KL(P_k \parallel M_k) + \frac{1}{2}KL(Q_k \parallel M_k)$.

and average across all $n$ samples in $D_{T_j}$. A symmetric computation is performed for samples from $D_{T_i}$. The final JSD-based task similarity score is:

$$\boldsymbol{S}_{\text{JSD}}(T_i; T_j) = \frac{1}{2}\left[\frac{1}{n}\sum_{k=1}^n \text{JSD}_k^{(j\leftarrow i)} + \frac{1}{m}\sum_{r=1}^m \text{JSD}_r^{(i\leftarrow j)}\right], \tag{5}$$

where each term quantifies the predictive distribution divergence when models are evaluated on out-of-task examples.

**Interpretability and Robustness** The Jensen-Shannon divergence provides several desirable properties in the context of task similarity: (i) symmetry under task permutation, (ii) boundedness within $[0, \log 2]$, which facilitates comparative analysis, and (iii) smooth behavior even when the support of distributions differ. Intuitively, low values of $\mathbf{S}_{\mathrm{JSD}}(T_i, T_j)$ suggest that the two tasks elicit similar probabilistic responses from their respective models—indicating potential overlap in learned structure, decision boundaries, or feature extraction routines. In contrast, high divergence implies task-specific specialization or misalignment in learned representations.

**Instance-Level Sampling Methodology** Upon getting an optimal probability mixture $\mathbf{p}^*$ over all $n$ tasks, $\mathbf{p}^*_{[i]}$ denoting the $i$-th task sampling probability, we define the samplewise selection over a multinomial distribution of the task wise mixture probabilities. We are given a total sampling budget of $B$ instances, and we wish to sample instances from the $n$ tasks such that the expected proportion of samples from task $i$ matches $p_i^*$.

To achieve this, we draw the task-wise instance counts $\mathbf{k} = [k_1, k_2, \ldots, k_n]$ from a multinomial distribution, where $\mathbf{k} \sim \mathrm{Multinomial}(B, \mathbf{p}^*)$

Each $k_i$ represents the number of instances to be drawn from task $i$. The probability mass function of the multinomial distribution is given by $P(k_1, \ldots, k_n; B, \mathbf{p}^*) = \frac{B!}{k_1! k_2! \ldots k_n!} \prod_{i=1}^{n} (p_i^*)^{k_i}$

## 5 TASK DISCOVERY

**Discrete Lifting of Continuous Mixture Optimization.** Given a current task submixture $\mathbf{\Pi}_k$ composed of $k$ tasks, our goal is to evaluate the marginal utility of introducing a candidate task $T_{k+1}$ to form an augmented mixture $\mathbf{\Pi}_{k+1}$. Let $V$ denote the universe of $n$ tasks, with $A \subseteq V$ indexing a subset and $\bar{A} \subseteq [n]$ denoting its corresponding index set. We define the continuous utility function over mixtures supported on $\bar{A}$ as

$$f(\bar{A}) := \max_{\mathbf{p} \in \Delta_n^{\mathbb{R}^+}; \, \mathrm{supp}(\mathbf{p}) \subseteq \bar{A}} \overline{\mathbb{E}}(\mathbf{p}) \tag{6}$$

where $\overline{\mathbb{E}}(\mathbf{p})$ denotes the negative objective of Eq 2). The maximizer over support set $\bar{A}$ is denoted by $\zeta^{(\bar{A})}$, so $f(\bar{A}) = \overline{\mathbb{E}}(\zeta^{(\bar{A})})$. To model incremental composition, we define the independent set family $I = \{S \subseteq V \mid |S| \le k\}$, and pose the top-$k$ task selection problem as $\max_{A \in I} f(\tilde{A})$, which lifts the relaxed optimization to a discrete set function defined over subsets of tasks. This formulation encourages incremental construction of $\mathbf{\Pi}_k$ by choosing the set $\bar{A}$ that supports the highest relaxed utility score under $\overline{\mathbb{E}}$.

> 🧪 (**Task Affinity**) : For mixtures $\mathbf{\Pi}_k$ and $\mathbf{\Pi}_{k+1}$ defined over the first $k$ and $k+1$ tasks respectively, let $\mathbf{p}_k$ and $\mathbf{p}_{k+1}$ be their corresponding mixture probability vectors. We define the *affinity* between these mixtures as the total variation (TV) distance between $\mathbf{p}_k$ and the marginalization of $\mathbf{p}_{k+1}$ over the first $k$ tasks, denoted $\mathbf{p}_{k+1}^{(k)}$:
>
> $$\mathrm{TV}(\mathbf{p}_k, \mathbf{p}_{k+1}^{(k)}) := \frac{1}{2} \sum_{i=1}^{k} |(\mathbf{p}_k)_i - (\mathbf{p}_{k+1})_i|.$$

This affinity measures the alignment between the task mixture before and after introducing the $(k+1)$-th task, with smaller values indicating higher consistency.

A **lower total variation divergence** indicates that the distribution over the first $k$ tasks remains stable when transitioning from the $k$-task mixture to the marginal of the $(k+1)$-task mixture. This stability reflects a strong affinity, demonstrating that the addition of the new task induces minimal perturbation to the existing task distribution.

## 6 THEORETICAL RESULTS

> **Lemma 1** (Monotonicity)**.** *Let $f$ be the set function defined in* (6)*. Then $f$ is monotonic: for any sets $\tilde{A} \subseteq \tilde{B}$, $f(\tilde{A}) \le f(\tilde{B})$.*

**Lemma 2.** *(Finite RSC and RSM) Let* $\mathbf{S} \in \mathbb{R}^{n \times n}$ *be a symmetric positive definite similarity matrix. Then the quadratic function* $\mathbb{E}(\mathbf{p}) = \mathbf{p}^\top \mathbf{S} \mathbf{p}$ *satisfies* Restricted Strong Convexity (RSC) *and* Restricted Smoothness (RSM) *over the probability simplex* $\Delta = \{\mathbf{p} \in \mathbb{R}^n : \mathbf{p} \geq 0, \|\mathbf{p}\|_1 = 1\}$ *with finite constants* $\mu > 0$ *and* $L > 0$, *respectively. That is, for all* $\mathbf{p}, \mathbf{q} \in \Delta$,

$$\frac{\mu}{2} \|\mathbf{p} - \mathbf{q}\|_2^2 \leq \mathbb{E}(\mathbf{p}) - \mathbb{E}(\mathbf{q}) - \nabla\mathbb{E}(\mathbf{q})^\top (\mathbf{p} - \mathbf{q}) \leq \frac{L}{2} \|\mathbf{p} - \mathbf{q}\|_2^2.$$

**Theorem 3.** *(Weak Submodularity) The set function* $f$ *in* (6) *is weakly submodular with the submodularity ratio* $\gamma > 0$.

## 7 EXPERIMENTAL SETUP

We evaluated several Instruction Fine-Tuning mixtures produced through our proposed probabilistic framework against several domain-specific knowledge and reasoning tasks as well as language understanding benchmarks, to comprehend and compare the fertility of the fine-tuned LLM. We show that applying our framework on a subset of large instruction tuning datasets, (1) LLMs fine-tuned on the derived mixture consistently out-perform heuristically sampled mixtures by at least 4% on MMLU and by more than 2% on some long context reasoning benchmarks from Open LLM Leaderboard; (2) low computation overheads on similarity matrix and mixture construction; 3) the correctness of our proposed algorithm and favorable properties of the similarity matrices were validated empirically to promote diversity and increase task representativeness.

**Models for Fine-Tuning** We evaluate TASKMIXPGM on **LLMs** ❶ *Llama-2-7B* (Touvron et al., 2023), ❷ *Mistral-7B-v0.3* (Jiang et al., 2023). We finetune the aforementioned models for one epoch on each dataset split, leveraging 8 NVIDIA H100 GPUs in bf16 precision. We use a per-device train batch size of 1, and using AdamW optimizer with a learning rate of $2 \times 10^{-5}$, weight decay 0.01, and gradient accumulation of 1 step. A linear learning-rate decay schedule is applied with a linear warmup over the first 3 % of total steps. To maximize memory efficiency, we enable gradient checkpointing and used DDP.

**Datasets for Submixtures** We evaluate our framework on a diverse set of instruction tuning datasets spanning language understanding and reasoning. These include: ❶ **Flan 2021** (Longpre et al., 2023; Chung et al., 2022), a multitask benchmark (~17.5M examples) aggregating prior datasets; ❷ **T0** (Sanh et al., 2021), an early prompt-driven multitask dataset for zero-shot generalization; ❸ **Chain-of-Thought (CoT)** (Wei et al., 2022), which augments prompts with intermediate steps to teach multi-step reasoning; ❹ **Tulu V3** (Lambert et al., 2024; Wang et al., 2023), a recent dataset with diverse, high-quality instructions from AI2; and ❺ **GLUE/SuperGLUE** (Wang et al., 2018; 2019), standard benchmarks for evaluating fine-grained language understanding and reasoning. These datasets collectively serve as a strong testbed for assessing our submixture selection method. In total we look at 319 tasks for creating our data mixture.

**Baselines for Comparison**: To show the efficacy of our proposed probabilistic framework, we compare against baselines which create mixtures heuristically, using some basic features of the tasks and combines them statistically and also introduces randomness in the overall process of constructing the mixture. For all experiments, we fix the hyperparameters controlling the balance between unary and pairwise terms, as well as the diversity penalty, i.e., the unary potential weight $\beta$ is set to 20, and the pairwise diversity penalty $\lambda$ is set to 10. We compare our methodology against 1) Uniform, which divides the total budget on the number of instances in the final mixture equally among all tasks and then samples the instances uniformly from each sub-task ; 2) EPM, splits total budget proportional to the number of instances in each sub-task, from which instances are sampled uniformly; 3) Random, sample the budget uniformly from the domain of all instances from all sub-tasks combined.

### 7.1 OBSERVATIONS

**PMI and JSD Perform Similarly Well**: We notice that that PMI-based selection consistently delivers superior performance, achieving the highest accuracy on MMLU for both Llama-2-7B and Mistral-7B, with improvements up to ~3–4% over uniform sampling and ~2–3% over random baselines. Since, PMI and JSD capture different aspects of similarity among tasks we notice that their relative performance lies within 1-2% showing very small divergence, hinting at a potential

choice of metric to be used for different objectives in a plug-and-play setting. Baseline methods exhibit competitive performance in isolated cases but lack consistency across datasets. The advantage of informed selection grows with dataset size, with PMI improving MMLU by ~3.5% on average from 25K to 100K samples, highlighting the scalability of principled task mixture design. These trends hold across both model families, underscoring the effectiveness of PMI and JSD for robust instruction-tuning.

Table 1: Llama2-7B: Instruct-tuning perf on MMLU and Leaderboard subsets with $\beta = 20, \lambda = 10$.

| | | MMLU | Leaderboard | | | | | |
|---|---|---|---|---|---|---|---|---|
| Dataset | Method | | BBH | GPQA | IFEval | Math | MMLU-Pro | MUSR |
| **25K** | | | | | | | | |
| 25K | Random | $0.3913_{\pm0.0040}$ | $0.3482_{\pm0.0059}$ | $0.2626_{\pm0.0128}$ | $\mathbf{0.3729}_{\pm N/A}$ | $0.0098_{\pm0.0027}$ | $\mathbf{0.1877}_{\pm0.0036}$ | $0.3677_{\pm0.0172}$ |
| 25K | Uniform | $0.3479_{\pm0.0039}$ | $0.3501_{\pm0.0059}$ | $0.2701_{\pm0.0129}$ | $0.3501_{\pm N/A}$ | $0.0151_{\pm0.0034}$ | $0.1768_{\pm0.0035}$ | $0.4127_{\pm0.0175}$ |
| 25K | EPM | $0.3802_{\pm0.0040}$ | $0.3593_{\pm0.0059}$ | $0.2601_{\pm0.0127}$ | $0.3405_{\pm N/A}$ | $0.0151_{\pm0.0033}$ | $0.1836_{\pm0.0035}$ | $\mathbf{0.4286}_{\pm0.0177}$ |
| 25K | Ours (PMI) | $\mathbf{0.4242}_{\pm0.0040}$ | $\mathbf{0.3598}_{\pm0.0059}$ | $0.2718_{\pm0.0129}$ | $0.3561_{\pm N/A}$ | $0.0136_{\pm0.0032}$ | $\mathbf{0.1877}_{\pm0.0036}$ | $0.4008_{\pm0.0174}$ |
| 25K | Ours (JSD) | $0.3926_{\pm0.0040}$ | $0.3454_{\pm0.0059}$ | $\mathbf{0.2785}_{\pm0.0130}$ | $0.3465_{\pm N/A}$ | $\mathbf{0.0151}_{\pm0.0034}$ | $0.1790_{\pm0.0035}$ | $0.4021_{\pm0.0175}$ |
| **50K** | | | | | | | | |
| 50K | Random | $0.4108_{\pm0.0040}$ | $0.3565_{\pm0.0060}$ | $0.2668_{\pm0.0128}$ | $0.3681_{\pm N/A}$ | $0.0144_{\pm0.0033}$ | $0.1881_{\pm0.0036}$ | $0.3770_{\pm0.0172}$ |
| 50K | Uniform | $0.3725_{\pm0.0040}$ | $0.3480_{\pm0.0059}$ | $0.2785_{\pm0.0130}$ | $\mathbf{0.4041}_{\pm N/A}$ | $0.0181_{\pm0.0037}$ | $0.1896_{\pm0.0036}$ | $\mathbf{0.4206}_{\pm0.0176}$ |
| 50K | EPM | $0.3801_{\pm0.0040}$ | $0.3532_{\pm0.0059}$ | $0.2634_{\pm0.0128}$ | $0.3507_{\pm N/A}$ | $0.0128_{\pm0.0031}$ | $0.1799_{\pm0.0035}$ | $0.4206_{\pm0.0176}$ |
| 50K | Ours (PMI) | $\mathbf{0.4156}_{\pm0.0040}$ | $0.3619_{\pm0.0060}$ | $0.2794_{\pm0.0130}$ | $0.3417_{\pm N/A}$ | $\mathbf{0.0189}_{\pm0.0037}$ | $0.1856_{\pm0.0035}$ | $0.3876_{\pm0.0174}$ |
| 50K | Ours (JSD) | $0.4074_{\pm0.0040}$ | $\mathbf{0.3624}_{\pm0.0060}$ | $\mathbf{0.2802}_{\pm0.0130}$ | $0.3525_{\pm N/A}$ | $0.0098_{\pm0.0027}$ | $\mathbf{0.1927}_{\pm0.0036}$ | $\mathbf{0.4206}_{\pm0.0176}$ |

Table 2: Mistral-7B: Instruct-tuning perf on MMLU and Leaderboard subsets with $\beta = 20, \lambda = 10$.

| | | MMLU | Leaderboard | | | | | |
|---|---|---|---|---|---|---|---|---|
| Dataset | Method | | BBH | GPQA | IFEval | Math | MMLU-Pro | MUSR |
| **50K** | | | | | | | | |
| 50K | Random | $0.4177_{\pm0.0040}$ | $0.3446_{\pm0.0059}$ | $0.2659_{\pm0.0128}$ | $0.4113_{\pm N/A}$ | $0.0106_{\pm0.0028}$ | $0.1733_{\pm0.0035}$ | $0.3836_{\pm0.0175}$ |
| 50K | Uniform | $\mathbf{0.4452}_{\pm0.0041}$ | $0.3479_{\pm0.0059}$ | $0.2651_{\pm0.0128}$ | $0.4161_{\pm N/A}$ | $0.0151_{\pm0.0033}$ | $0.1799_{\pm0.0035}$ | $0.3823_{\pm0.0172}$ |
| 50K | EPM | $0.4405_{\pm0.0041}$ | $0.3413_{\pm0.0059}$ | $0.2701_{\pm0.0129}$ | $\mathbf{0.4293}_{\pm N/A}$ | $0.0174_{\pm0.0036}$ | $0.1871_{\pm0.0036}$ | $0.4034_{\pm0.0174}$ |
| 50K | Ours (PMI) | $0.4228_{\pm0.0040}$ | $0.3492_{\pm0.0058}$ | $\mathbf{0.2735}_{\pm0.0129}$ | $0.3094_{\pm N/A}$ | $\mathbf{0.0174}_{\pm0.0036}$ | $0.1758_{\pm0.0035}$ | $\mathbf{0.4259}_{\pm0.0176}$ |
| 50K | Ours (JSD) | $0.4138_{\pm0.0040}$ | $\mathbf{0.3498}_{\pm0.0059}$ | $0.2567_{\pm0.0127}$ | $0.4065_{\pm N/A}$ | $0.0159_{\pm0.0034}$ | $\mathbf{0.1898}_{\pm0.0035}$ | $0.3890_{\pm0.0173}$ |
| **100K** | | | | | | | | |
| 100K | Random | $0.4476_{\pm0.0041}$ | $0.3416_{\pm0.0060}$ | $0.2542_{\pm0.0126}$ | $\mathbf{0.4388}_{\pm N/A}$ | $0.0186_{\pm0.0038}$ | $0.1730_{\pm0.0034}$ | $0.4048_{\pm0.0175}$ |
| 100K | Uniform | $0.4486_{\pm0.0041}$ | $0.3532_{\pm0.0059}$ | $\mathbf{0.2661}_{\pm0.0128}$ | $0.3741_{\pm N/A}$ | $0.0174_{\pm0.0036}$ | $0.1724_{\pm0.0034}$ | $0.3810_{\pm0.0173}$ |
| 100K | EPM | $0.4505_{\pm0.0041}$ | $0.3578_{\pm0.0060}$ | $0.2466_{\pm0.0125}$ | $\mathbf{0.4388}_{\pm N/A}$ | $0.0174_{\pm0.0036}$ | $0.1859_{\pm0.0035}$ | $0.4074_{\pm0.0175}$ |
| 100K | Ours (PMI) | $\mathbf{0.5476}_{\pm0.0040}$ | $0.3388_{\pm0.0058}$ | $0.2508_{\pm0.0126}$ | $0.3369_{\pm N/A}$ | $0.0136_{\pm0.0032}$ | $0.1810_{\pm0.0035}$ | $0.4081_{\pm0.0176}$ |
| 100K | Ours (JSD) | $0.5301_{\pm0.0040}$ | $\mathbf{0.3591}_{\pm0.0060}$ | $0.2667_{\pm0.0127}$ | $0.4137_{\pm N/A}$ | $\mathbf{0.0189}_{\pm0.0037}$ | $\mathbf{0.1953}_{\pm0.0035}$ | $\mathbf{0.4140}_{\pm0.0175}$ |

| | highest accuracy | | 2nd highest accuracy | | 3rd highest accuracy. |
|---|---|---|---|---|---|

**More Samples Boost Performance on Complex Benchmarks**: Increasing the number of instances in the mixtures from 25K to 50K and to 100K, reflects in improved performance in MMLU and MUSR, which requires complex skills such as long-context reasoning and language understanding and 10+% increase in accuracy on MMLU with Mistral-7B with PMI as well as in JSD, though we see higher accuracy than other heuristics driven methods with half the samples. In general, we observe that we consistently perform better than the baselines on basic and graduate level mathematical reasoning tasks(MMLU, MMLU-Pro), language and reasoning tasks(BBH, MUSR) and other domain knowledge tasks(GPQA), proving the effectiveness of our simple probabilistic framework.

**Uniform and EPM Fail to Generalize**: Although Uniform and EPM achieve competitive results in isolated cases, their overall performance in Table 1 and 2 [7.1] reveals weak generalization across benchmarks and scales. For instance, Uniform achieves the best MMLU accuracy at 50K on Mistral-7B (0.4452), yet its average gain across datasets is only 1.0% over random sampling, compared to 3.5% for PMI. EPM similarly excels in narrow settings, such as MUSR at 25K on Llama2-7B (0.4286), but fails on reasoning-intensive benchmarks like GPQA or MMLU-Pro, often trailing the random baseline. This fragmented behavior suggests reliance on superficial correlations rather than robust task relevance. In contrast, PMI and JSD-based selection not only achieve higher peaks but also remain stable across dataset sizes (25K–100K) and model families, underscoring the need for principled similarity metrics in scalable instruction tuning.

## 7.2 ABLATION STUDIES

To better understand the impact of different similarity metrics on the structure of the similarity matrices, we analyze the eigenvalue spectra of matrices computed using Jensen-Shannon Divergence (JSD) and Pointwise Mutual Information (PMI). Figure 1 presents the sorted eigenvalues, revealing distinct spectral decay patterns for the two metrics. The sharper decay observed in the PMI-based similarity matrix (Figure 1b) suggests a lower effective rank, which corresponds to a more concentrated representation of inter-sample relationships. In contrast, the JSD-based matrix (Figure 1a) exhibits a more gradual decay, indicating a richer but potentially noisier similarity structure.

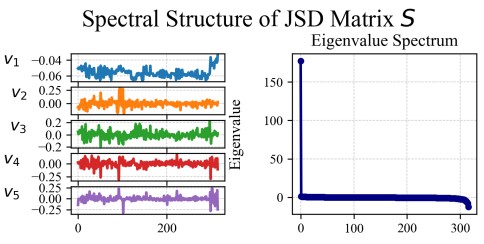 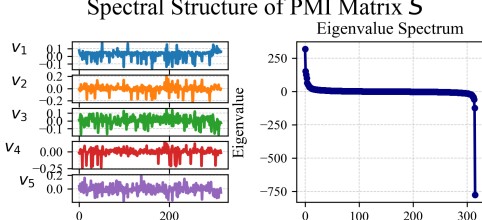

(a) JSD Similarity Matrix Eigenvalue Spectrum    (b) PMI Similarity Matrix Eigenvalue Spectrum

Figure 1: **Eigenvalue spectra** of similarity matrices derived from **(a) Jensen-Shannon Divergence (JSD)** and **(b) Pointwise Mutual Information (PMI)**. The **PMI-based matrix** exhibits a **steeper spectral decay**, indicating a **lower effective rank** and thus a **more compact embedding** of similarity relationships.

**Task Discovery.** We study how adding new tasks to an existing mixture $\Pi_k$ affects the distribution, focusing on mass redistribution and the utility of the new task. We analyze two scenarios: (i) adding tasks in descending order of unary potential $\beta \mathbf{S}_i$, and (ii) in ascending order. This helps characterize the influence of strong versus weak unary potentials on the optimized mixture and whether high-unary tasks dominate or reinforce existing clusters.

## 8 CONCLUSION

We presented **TASKMIXPGM**, a theoretically grounded framework for optimizing fine-tuning task mixtures in large language models. By modeling task relationships as an energy minimization over an MRF, **TASKMIXPGM** derives closed-form optimal task proportions that balance utility and diversity. Unlike prior heuristics, it leverages output distribution divergences to capture functional task behavior. Our experiments shows consistent improvements over Uniform, Random, and EPM, with PMI and JSD method. Ablations confirm the importance of spectral correction and hyperparameter stability, providing both theoretical and practical robustness. While similarity estimation adds overhead, it produces reusable mixtures across budgets, making the upfront cost worthwhile.

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
