# Supplementary Material: TaskMixPGM: Task Mixtures via Probabilistic Graphical Modelling for Language Model Finetuning

## CONTENTS

## Supplementary Material: TaskMixPGM: Task Mixtures via Probabilistic Graphical Modelling for Language Model Finetuning

## A  ORGANIZATION OF THE APPENDIX

This appendix provides supporting material for the main text, organized into the following sections. Section B presents the overall broader impact of our work. Section C presents the theoretical foundations underpinning our approach, including monotonicity and submodularity results relevant to energy-based models. Section D provides a comparative analysis of task similarity measures, starting with linearized fine-tuning vectors and extending to distributional metrics such as Pointwise Mutual Information (PMI) and Jensen-Shannon Divergence (JSD), along with algorithms for their computation. Section E details the experimental setup, datasets, and model configurations used in our evaluations. Section F includes extended results, such as tabular comparisons, that complement those in the main paper. Finally, Section G outlines the structure of our codebase and provides guidance for reproducing the experiments.

## B  BROADER IMPACT

Our proposed work on TASKMIXPGM has significant broader impact across multiple domains of machine learning research and real-world applications.

- In **natural language understanding and multilingual benchmarks**, the selection of fine-tuning data mixtures is critical to model generalization. By explicitly optimizing for both representativeness and diversity, TASKMIXPGM enhances performance on complex, multi-domain evaluations such as MMLU and BIG-Bench-Hard. This enables more robust LLMs capable of reasoning across languages, topics, and task formats.

- In **AI deployment for low-resource and specialized domains**, TASKMIXPGM provides a scalable and principled solution to constructing effective mixtures from limited or domain-specific task collections. Applications include legal document analysis, medical QA, and scientific literature synthesis—areas where manually tuning mixtures is costly and error-prone.

- In **AI safety and interpretability** research, our framework offers interpretable insights into task interactions and data influence. The use of functional similarity via output divergences, rather than opaque semantic features, facilitates transparency in fine-tuning decisions. This can assist auditing pipelines and mitigate risks associated with over-representation of narrow task distributions.

- In **efficient model training and green AI initiatives**, TASKMIXPGM can reduce unnecessary computation and data usage by guiding mixture construction toward high-impact tasks. This aligns with ongoing efforts to lower the carbon footprint of large-scale model development while maintaining or improving downstream performance.

## C  MAIN THEORETICAL RESULTS

### C.1  CLOSED-FORM SOLUTION OF QUADRATIC MINIMIZATION OVER THE SIMPLEX

We consider the problem of minimizing a quadratic energy function over the probability simplex $\Delta_n = \{\mathbf{p} \in \mathbb{R}^n : \mathbf{p}^\top \mathbf{1}_n = 1, \ \mathbf{p} \geq 0\}$:

$$\min_{\mathbf{p} \in \Delta_n} \ E(\mathbf{p}) := -\boldsymbol{\Psi}_{\text{un}}^\top \mathbf{p} + \frac{1}{2}\mathbf{p}^\top \boldsymbol{\Psi}_{\text{pair}}\mathbf{p} \tag{7}$$

where $\boldsymbol{\Psi}_{\text{un}} \in \mathbb{R}^n$ denotes a unary potential vector and $\boldsymbol{\Psi}_{\text{pair}} \in \mathbb{R}^{n \times n}$ is a symmetric positive semi-definite (PSD) matrix encoding pairwise interactions.

## C.2 LAGRANGIAN AND FIRST-ORDER CONDITIONS

To enforce the affine constraint $\mathbf{p}^\top \mathbf{1}_n = 1$, and inequality constraints $\mathbf{p} \geq 0$, we consider the KKT conditions for optimality. Define the Lagrangian:

$$L(\mathbf{p}, \nu, \boldsymbol{\mu}) = -\boldsymbol{\Psi}_{\text{un}}^\top \mathbf{p} + \frac{1}{2}\mathbf{p}^\top \boldsymbol{\Psi}_{\text{pair}}\mathbf{p} + \nu(\mathbf{p}^\top \mathbf{1}_n - 1) - \boldsymbol{\mu}^\top \mathbf{p} \tag{8}$$

with dual variables $\nu \in \mathbb{R}$ (equality) and $\boldsymbol{\mu} \in \mathbb{R}_+^n$ (inequality).

The **KKT optimality conditions** are:

Stationarity Condition

$$\nabla_{\mathbf{p}} L = -\boldsymbol{\Psi}_{\text{un}} + \boldsymbol{\Psi}_{\text{pair}}\mathbf{p} + \nu\mathbf{1}_n - \boldsymbol{\mu} = 0$$

$$\frac{\partial L}{\partial \mathbf{p}_{[i]}} = -\boldsymbol{\Psi}_{\text{un}[i]} + \sum_{j=1}^{n} \boldsymbol{\Psi}_{\text{pair}[ij]}\mathbf{p}_{[j]} + \nu - \boldsymbol{\mu}_{[i]} = 0.$$

This implies:

$$\nu = \boldsymbol{\Psi}_{\text{un}[i]} - \sum_{j=1}^{n} \boldsymbol{\Psi}_{\text{pair}[ij]}\,\mathbf{p}_{[j]} + \boldsymbol{\mu}_{[i]}.$$

Primal Feasibility

$$0 \leq \mathbf{p}_{[i]} \leq 1, \quad \text{for } i = 1, \ldots, n, \quad \sum_{i=1}^{n} \mathbf{p}_{[i]} = 1$$

Dual Feasibility

$$\nu \in \mathbb{R}^n, \quad \boldsymbol{\mu}_{[i]} \geq 0$$

Complementary Slackness

$$\boldsymbol{\mu}_{[i]}\mathbf{p}_{[i]} = 0 \quad \forall i \in [n]$$

Coordinate wise analysis for each edge cases

- (**Interior Points**) $0 < \mathbf{p}_{[i]} < 1$ Due to complementary slackness, we have $\boldsymbol{\mu}_{[i]} = 0 \ \forall i \in [n]$. Hence $\nu = \boldsymbol{\Psi}_{\text{un}[i]} - \sum_{j=1}^{n} \boldsymbol{\Psi}_{\text{pair}[ij]}\,\mathbf{p}_{[j]}$ and therefore $\quad \mathbf{p}_{[i]} = \sum_{j=1}^{n} \boldsymbol{\Psi}_{\text{pair}[ij]}^{-1}(\boldsymbol{\Psi}_{\text{un}} - \nu\mathbf{1}_n)$

- (**Boundary Point**) $\mathbf{p}_{[i]} = 0$ From complementary slackness, we know $\boldsymbol{\mu}_{[i]} \geq 0$

Let $k$ points lie in the interior and $n - k$ points lie on the boundary

$$\sum_{i \in k} e^{\frac{\mathbb{S}_a[i] - \beta}{\lambda} - 1} + \sum_{i \in (n-k)} \mathbf{0}_{[i]} = 1$$

## C.3 SOLUTION UNDER INTERIOR ASSUMPTION

We first consider the case where the solution lies in the relative interior of the simplex; that is, $\mathbf{p}^* > 0$ and hence $\boldsymbol{\mu} = \mathbf{0}$. Substituting into (**??**), we obtain:

$$\boldsymbol{\Psi}_{\text{pair}}\mathbf{p} = \boldsymbol{\Psi}_{\text{un}} - \nu\mathbf{1}_n \tag{9}$$

Assuming $\boldsymbol{\Psi}_{\text{pair}}$ is invertible (i.e., strictly positive definite), we may solve:

$$\mathbf{p} = \boldsymbol{\Psi}_{\text{pair}}^{-1}\boldsymbol{\Psi}_{\text{un}} - \nu\boldsymbol{\Psi}_{\text{pair}}^{-1}\mathbf{1}_n \tag{10}$$

Imposing the constraint $\mathbf{p}^\top \mathbf{1}_n = 1$, we find:

$$\mathbf{1}_n^\top\mathbf{p} = \mathbf{1}_n^\top \boldsymbol{\Psi}_{\text{pair}}^{-1}\boldsymbol{\Psi}_{\text{un}} - \nu\mathbf{1}_n^\top \boldsymbol{\Psi}_{\text{pair}}^{-1}\mathbf{1}_n = 1 \tag{11}$$

Letting

$$a := \mathbf{1}_n^\top \mathbf{\Psi}_{\text{pair}}^{-1} \mathbf{\Psi}_{\text{un}}, \quad b := \mathbf{1}_n^\top \mathbf{\Psi}_{\text{pair}}^{-1} \mathbf{1}_n,$$

we obtain $\nu = \frac{a-1}{b}$.

Substituting back into the expression for $\mathbf{p}$, we conclude:

$$\boxed{\mathbf{p}^* = \mathbf{\Psi}_{\text{pair}}^{-1} \mathbf{\Psi}_{\text{un}} - \frac{\mathbf{1}_n^\top \mathbf{\Psi}_{\text{pair}}^{-1} \mathbf{\Psi}_{\text{un}} - 1}{\mathbf{1}_n^\top \mathbf{\Psi}_{\text{pair}}^{-1} \mathbf{1}_n} \cdot \mathbf{\Psi}_{\text{pair}}^{-1} \mathbf{1}_n} \tag{12}$$

## C.4 DISCUSSION

The closed-form expression (12) satisfies the affine constraint by construction. If $\mathbf{p}^* \geq 0$ componentwise, it is the unique global minimizer. Otherwise, if any coordinate is negative, the interior assumption fails, and active-set refinement or projection onto the simplex is required. In practice, one may use projection-based algorithms (e.g., conditional gradient, projected gradient descent) or iteratively restrict to the support set of nonnegative entries and resolve (12) over that face of the simplex.

## C.5 MONOTONICITY AND SUBMODULAR PROPERTIES OF ENERGY POTENTIAL

**Lemma 1** (Monotonicity). *Let $f$ be the set function defined in Eq (4). Then $f$ is monotonic: for any sets $\tilde{A} \subseteq \tilde{B}$, $f(\tilde{A}) \leq f(\tilde{B})$.*

*Proof.* Let $|\tilde{A}| = n_1$ and $|\tilde{B}| = n_2$ and since $\tilde{A} \subseteq \tilde{B}$ we have $n_1 < n_2$. We index the elements in $\tilde{B}$ such that the first $n_1$ elements are contained in $\tilde{A}$.

$$f(\tilde{B}) = \max_{\mathbf{p} \in \Delta_{n_2}^{\mathbb{R}}; \, \text{supp}(\mathbf{p}) \subseteq \bar{B}} \overline{\mathbb{E}}(\mathbf{p}) \geq \max_{\mathbf{p} \in \Delta_{n_1}^{\mathbb{R}}; \, \text{supp}(\mathbf{p}) \subseteq \bar{A}} \overline{\mathbb{E}}(\mathbf{p}) = f(\tilde{A})$$

$\square$

This indicates the function under consideration is monotonically increasing under task mixture.

**Lemma 2** (Finite RSC and RSM of Quadratic Term). *Let $\mathbf{S} \in \mathbb{R}^{n \times n}$ be a symmetric positive definite similarity matrix. Then the quadratic function $\mathbb{E}(\mathbf{p}) = \mathbf{p}^\top \mathbf{S} \mathbf{p}$ satisfies* Restricted Strong Convexity *(RSC) and* Restricted Smoothness *(RSM) over the probability simplex $\Delta_n = \{\mathbf{p} \in \mathbb{R}^n : \mathbf{p} \geq 0, \|\mathbf{p}\|_1 = 1\}$ with finite constants $c_\Omega > 0$ and $C_\Omega > 0$, respectively. That is, for all $\mathbf{p}, \mathbf{q} \in \Delta_n$,*

$$\frac{c_\Omega}{2} \|\mathbf{p} - \mathbf{q}\|_2^2 \leq \mathbb{E}(\mathbf{p}) - \mathbb{E}(\mathbf{q}) - \nabla \mathbb{E}(\mathbf{q})^\top (\mathbf{p} - \mathbf{q}) \leq \frac{C_\Omega}{2} \|\mathbf{p} - \mathbf{q}\|_2^2.$$

*Proof.* Let $\mathbb{E}(\mathbf{p}) := \mathbf{p}^\top \mathbf{S} \mathbf{p}$ denote the energy of the task mixture $\mathbf{p} \in \Delta$, where $\mathbf{S} \in \mathbb{R}^{n \times n}$ is a symmetric positive definite similarity matrix and $n$ denotes the total number of tasks. We may express the second-order Taylor expansion of $\mathbb{E}$ as:

$$\mathbb{E}(\mathbf{p}) = \mathbb{E}(\mathbf{q}) + \nabla \mathbb{E}(\mathbf{q})^\top (\mathbf{p} - \mathbf{q}) + \frac{1}{2} (\mathbf{p} - \mathbf{q})^\top \nabla^2 \mathbb{E}(\xi)(\mathbf{p} - \mathbf{q})$$

for some $\xi$ on the line segment between $\mathbf{p}$ and $\mathbf{q}$.

Since $\nabla \mathbb{E}(\mathbf{p}) = 2\mathbf{S}\mathbf{p}$ and $\nabla^2 \mathbb{E}(\mathbf{p}) = 2\mathbf{S}$ is constant over $\mathbf{p}$, we simplify the residual energy term:

$$\mathbb{E}(\mathbf{p}) - \mathbb{E}(\mathbf{q}) - \nabla \mathbb{E}(\mathbf{q})^\top (\mathbf{p} - \mathbf{q}) = (\mathbf{p} - \mathbf{q})^\top \mathbf{S} (\mathbf{p} - \mathbf{q})$$

We now invoke spectral bounds on the quadratic form. Let $\lambda_{\min}(\mathbf{S}), \lambda_{\max}(\mathbf{S})$ denote the smallest and largest eigenvalues of $\mathbf{S}$. Since $\mathbf{S} > 0$, we have:

$$\lambda_{\min}(\mathbf{S}) \|\mathbf{p} - \mathbf{q}\|_2^2 \leq (\mathbf{p} - \mathbf{q})^\top \mathbf{S} (\mathbf{p} - \mathbf{q}) \leq \lambda_{\max}(\mathbf{S}) \|\mathbf{p} - \mathbf{q}\|_2^2$$

Combining with the expression above, we obtain the sandwich bound:

$$\lambda_{\min}(\mathbf{S}) \|\mathbf{p} - \mathbf{q}\|_2^2 \leq \mathbb{E}(\mathbf{p}) - \mathbb{E}(\mathbf{q}) - \nabla \mathbb{E}(\mathbf{q})^\top (\mathbf{p} - \mathbf{q}) \leq \lambda_{\max}(\mathbf{S}) \|\mathbf{p} - \mathbf{q}\|_2^2$$

Defining $c_\Omega \coloneqq 2\lambda_{\min}(\mathbf{S})$ and $L \coloneqq 2\lambda_{\max}(\mathbf{S})$, we conclude that $\mathbb{E}(\mathbf{p})$ is $(c_\Omega, C_\Omega)$-restricted strongly convex and smooth over $\Delta$ in the sense that:

$$\frac{c_\Omega}{2}\|\mathbf{p} - \mathbf{q}\|_2^2 \leq \mathbb{E}(\mathbf{p}) - \mathbb{E}(\mathbf{q}) - \nabla\mathbb{E}(\mathbf{q})^\top(\mathbf{p} - \mathbf{q}) \leq \frac{C_\Omega}{2}\|\mathbf{p} - \mathbf{q}\|_2^2$$

$\square$

---

**Lemma 3** (Finite RSC and RSM of Eq: 1 Energy Potential). *Let $\mathbf{S} \in \mathbb{R}^{n \times n}$ be a symmetric positive definite similarity matrix. Then the quadratic function $\mathbb{E}(\mathbf{p}) = -\mathbf{\Psi}_{un}^\top\mathbf{p} + \frac{1}{2}\mathbf{p}^\top\mathbf{\Psi}_{pair}\mathbf{p}$ satisfies* Restricted Strong Convexity (RSC) *with parameter $c_\Omega$ and* Restricted Smoothness (RSM) *with parameter $C_\Omega$ over the probability simplex $\Delta_n = \{\mathbf{p} \in \mathbb{R}^n : \mathbf{p} \geq 0, \|\mathbf{p}\|_1 = 1\}$ with finite constants $c_\Omega > 0$ and $C_\Omega > 0$, respectively. That is, for all $\mathbf{p}, \mathbf{q} \in \Delta_n$,*

$$\frac{c_\Omega}{2}\|\mathbf{p} - \mathbf{q}\|_2^2 \leq \mathbb{E}(\mathbf{p}) - \mathbb{E}(\mathbf{q}) - \nabla\mathbb{E}(\mathbf{q})^\top(\mathbf{p} - \mathbf{q}) \leq \frac{C_\Omega}{2}\|\mathbf{p} - \mathbf{q}\|_2^2.$$

---

*Proof.* We begin by analyzing the structure of the energy function $\mathbb{E} : \mathbb{R}^n \to \mathbb{R}$, defined as

$$\mathbb{E}(\mathbf{p}) = -\mathbf{\Psi}_{un}^\top\mathbf{p} + \frac{1}{2}\mathbf{p}^\top\mathbf{\Psi}_{pair}\mathbf{p}.$$

This function is a standard quadratic form, with gradient and Hessian given by

$$\nabla\mathbb{E}(\mathbf{p}) = \mathbf{\Psi}_{pair}\mathbf{p} - \mathbf{\Psi}_{un}, \quad \nabla^2\mathbb{E}(\mathbf{p}) = \mathbf{\Psi}_{pair}.$$

Since $\mathbf{\Psi}_{pair}$ is symmetric positive definite, it admits an eigenvalue decomposition $\mathbf{\Psi}_{pair} = \mathbf{U}\mathbf{\Lambda}\mathbf{U}^\top$ with eigenvalues $0 < \lambda_1 \leq \cdots \leq \lambda_n$. Let $c_\Omega \coloneqq \lambda_{\min}(\mathbf{\Psi}_{pair})$ and $C_\Omega \coloneqq \lambda_{\max}(\mathbf{\Psi}_{pair})$.

We now apply the standard second-order Taylor expansion of $\mathbb{E}$ at $\mathbf{q} \in \Delta$ evaluated at $\mathbf{p} \in \Delta$:

$$\mathbb{E}(\mathbf{p}) = \mathbb{E}(\mathbf{q}) + \nabla\mathbb{E}(\mathbf{q})^\top(\mathbf{p} - \mathbf{q}) + \frac{1}{2}(\mathbf{p} - \mathbf{q})^\top\mathbf{\Psi}_{pair}(\mathbf{p} - \mathbf{q}),$$

and hence,

$$\mathbb{E}(\mathbf{p}) - \mathbb{E}(\mathbf{q}) - \nabla\mathbb{E}(\mathbf{q})^\top(\mathbf{p} - \mathbf{q}) = \frac{1}{2}(\mathbf{p} - \mathbf{q})^\top\mathbf{\Psi}_{pair}(\mathbf{p} - \mathbf{q}).$$

Applying the Rayleigh quotient bounds for the positive definite matrix $\mathbf{\Psi}_{pair}$, we obtain

$$c_\Omega\|\mathbf{p} - \mathbf{q}\|_2^2 \leq (\mathbf{p} - \mathbf{q})^\top\mathbf{\Psi}_{pair}(\mathbf{p} - \mathbf{q}) \leq C_\Omega\|\mathbf{p} - \mathbf{q}\|_2^2,$$

and thus

$$\frac{c_\Omega}{2}\|\mathbf{p} - \mathbf{q}\|_2^2 \leq \mathbb{E}(\mathbf{p}) - \mathbb{E}(\mathbf{q}) - \nabla\mathbb{E}(\mathbf{q})^\top(\mathbf{p} - \mathbf{q}) \leq \frac{C_\Omega}{2}\|\mathbf{p} - \mathbf{q}\|_2^2.$$

This establishes that $\mathbb{E}$ is $c_\Omega$-strongly convex and $C_\Omega$-smooth over the probability simplex $\Delta$, with constants determined by the minimal and maximal eigenvalues of $\mathbf{\Psi}_{pair}$. $\square$

*Note*: In any case even if $\mathbf{\Psi}_{pair}$ is non-psd, psd correction via Spectral Shifting can be utilised to make it a psd matrix.

## C.6 WEAK SUBMODULARITY OF SET FUNCTION $f$

**Theorem 1.** *(Weak Submodularity) The set function $f(\tilde{A}) \coloneqq \max_{\mathbf{p} \in \Delta_{n_1}^\mathbb{R}; \ \mathrm{supp}(\mathbf{p}) \subseteq \bar{A}} \overline{\mathbb{E}}(\mathbf{p})$ in Eq (4) is weakly submodular where $\mathbb{E}(\mathbf{p}) = -\mathbf{\Psi}_{un}^\top\mathbf{p} + \frac{1}{2}\mathbf{p}^\top\mathbf{\Psi}_{pair}\mathbf{p}$ with the submodularity ratio $\gamma > 0$.*

*Proof.* Let $L, S \subseteq [n_1]$ be disjoint sets and define $m = |L| + |S|$. Let $\zeta(L) = \arg\max_{\mathbf{p} \in \Delta^\mathbb{R}, \ \mathrm{supp}(\mathbf{p}) \subseteq L} \mathbb{E}(\mathbf{p})$ and similarly define $\zeta(L \cup S)$ for the superset.

By the Restricted Strong Convexity (RSC) and Restricted Smoothness (RSM) of $\mathbb{E}$ over the probability simplex (proved previously), we have for constants $c_\Omega > 0$, $C_\Omega > 0$, and for any $\mathbf{p}, \mathbf{q}$ supported in a set of size $m$,

$$\frac{c_\Omega}{2}\|\mathbf{p} - \mathbf{q}\|_2^2 \leq \mathbb{E}(\mathbf{p}) - \mathbb{E}(\mathbf{q}) - \nabla\mathbb{E}(\mathbf{q})^\top(\mathbf{p} - \mathbf{q}) \leq \frac{C_\Omega}{2}\|\mathbf{p} - \mathbf{q}\|_2^2.$$

Let us upper bound the total gain from adding $S$ to $L$:

$$f(L \cup S) - f(L) = \mathbb{E}(\zeta(L \cup S)) - \mathbb{E}(\zeta(L)).$$

By the descent lemma and RSM,

$$\mathbb{E}(\zeta(L \cup S)) - \mathbb{E}(\zeta(L)) \leq \langle \nabla \mathbb{E}(\zeta(L)), \zeta(L \cup S) - \zeta(L) \rangle - \frac{c_\Omega}{2} \|\zeta(L \cup S) - \zeta(L)\|^2.$$

We upper bound the inner product using the point $\mathbf{v}$ defined as the projected optimal update within the support $L \cup S$. That is,

$$v_{L \cup S} = \max \left\{ \frac{1}{c_\Omega} \nabla \mathbb{E}_{L \cup S}(\zeta(L)) + \zeta(L)_{L \cup S}, \ 0 \right\}.$$

Since $\zeta(L \cup S)$ maximizes $\mathbb{E}$ over support $L \cup S$, and $\mathbf{v}$ is a feasible direction, we can use:

$$\mathbb{E}(\zeta(L \cup S)) - \mathbb{E}(\zeta(L)) \leq \langle \nabla \mathbb{E}(\zeta(L)), \mathbf{v} - \zeta(L) \rangle - \frac{c_\Omega}{2} \|\mathbf{v} - \zeta(L)\|^2.$$

Now consider the coordinate-wise marginal gains. For each $j \in S$, we define the directional gain from adding $j$ to $L$ as:

$$f(L \cup \{j\}) - f(L) \geq \max_{\alpha \geq 0} \left[ \langle \nabla_j \mathbb{E}(\zeta(L)), \alpha \rangle - \frac{L}{2} \alpha^2 \right] = \frac{1}{2 C_\Omega} [\nabla_j \mathbb{E}(\zeta(L))]_+^2.$$

Summing over $j \in S$ where $\nabla_j \mathbb{E}(\zeta(L)) > 0$, we get

$$\sum_{j \in S} f(L \cup \{j\}) - f(L) \geq \frac{1}{2 C_\Omega} \|\nabla_S^+ \mathbb{E}(\zeta(L))\|^2.$$

From the earlier upper bound, we had

$$f(L \cup S) - f(L) \leq \langle \nabla \mathbb{E}(\zeta(L)), \mathbf{v} - \zeta(L) \rangle - \frac{c_\Omega}{2} \|\mathbf{v} - \zeta(L)\|^2.$$

The maximizer of this expression occurs at:

$$v_j = \max \left\{ \frac{1}{c_\Omega} \nabla_j \mathbb{E}(\zeta(L)), \ 0 \right\}.$$

This gives:

$$f(L \cup S) - f(L) \leq \frac{1}{2 c_\Omega} \|\nabla_S^+ \mathbb{E}(\zeta(L))\|^2.$$

Combining the lower and upper bounds:

$$\sum_{j \in S} f(L \cup \{j\}) - f(L) \geq \frac{1}{2 C_\Omega} \|\nabla_S^+ \mathbb{E}(\zeta(L))\|^2, \quad f(L \cup S) - f(L) \leq \frac{1}{2 c_\Omega} \|\nabla_S^+ \mathbb{E}(\zeta(L))\|^2.$$

Hence,

$$\sum_{j \in S} f(L \cup \{j\}) - f(L) \geq \frac{c_\Omega}{C_\Omega} \left( f(L \cup S) - f(L) \right),$$

which proves weak submodularity with submodularity ratio $\gamma = c_\Omega / C_\Omega > 0$. $\qquad \square$

# D  COMPARATIVE ANALYSIS ACROSS VARIOUS NOTIONS OF TASK SIMILARITIES

## D.1  SIMILARITY ACROSS TASK VECTORS VIA LINEARIZED FINETUNING

Large-scale pretrained language models (PLMs) such as GPT-2 are widely adapted to downstream tasks via full-model fine-tuning. However, multi-task or per-task retraining remains computationally burdensome. *Task arithmetic* **?** introduces a simple yet effective approach: given a pretrained checkpoint initialization $\boldsymbol{\theta}_0$ and task-specific fine-tuned weights $\boldsymbol{\theta}_t^*$, the *task vector* is defined as:

$$\tau_t := \boldsymbol{\theta}_t^* - \boldsymbol{\theta}_0$$

These vectors enable model editing via linear composition:

- **Addition:** $\boldsymbol{\theta}_0 + \sum_{t \in T} \tau_t$ synthesizes multi-task behaviors.

- **Negation:** $\boldsymbol{\theta}_0 - \tau_s$ induces task-specific forgetting.

While effective, the underlying mechanisms behind this arithmetic remain poorly understood.

**Linearized Fine-Tuning**: (**?**) posit that *tangent-space fine-tuning* disentangles task behaviors more effectively by constraining updates to the local linear approximation of the model. Let $f(x; \boldsymbol{\theta})$ denote a PLM with parameters $\boldsymbol{\theta} \in \mathbb{R}^m$, the corresponding **nonlinear task vector** is given by $\tau_t^{\text{nl}} := \boldsymbol{\theta}_t^* - \boldsymbol{\theta}_0$.

In contrast, *linearized fine-tuning* restricts optimization to the first-order Taylor expansion:

$$f_{\text{lin}}(x; \boldsymbol{\theta}) := f(x; \boldsymbol{\theta}_0) + \nabla_{\boldsymbol{\theta}} f(x; \boldsymbol{\theta}_0)^\top (\boldsymbol{\theta} - \boldsymbol{\theta}_0)$$

This surrogate is optimized using Jacobian-vector products (JVP), yielding a linearized task vector:

$$\tau_t^{\text{lin}} := \boldsymbol{\theta}_t^{\text{lin}*} - \boldsymbol{\theta}_0$$

Task vectors are generally useful as they can enable model editing as well provide a well defined representation of the finetuning task at hand, dependent on the model parameters. Ideally, the goal would be to select multiple linearly independent task vectors such that they represent generalizably well across a range of IFT datasets and does generalizably well across different benchmark datasets. The algorithm is presented as Algorithm 1 in Section D.2.

**Similarity Structure of Task Embeddings**

Directly computing any similarity metric over $m \sim 10^6$ to $10^9$ parameters, is computationally expensive. Thus, we first isolate the most informative layer (chosen via task-vector analysis using **layer-wise subsetting** and then project its high-dimensional slice task vector $\tau \in \mathbb{R}^m$ to a much lower-dimensional vector $\tilde{\tau} = R\tau \in \mathbb{R}^k$ using a **Gaussian random matrix** $R \in \mathbb{R}^{k \times m}$ with $k \ll m$. This projection technique is known to preserve similarity distances in expectation, providing a reliable and efficient approximation for comparing vector directions in the reduced space.

**Cosine Similarity across Task Vectors**: To analyze inter-task relationships, we examine the cosine similarity between task vectors:

$$\text{sim}(\tau_A, \tau_B) := \frac{\tau_A^\top \tau_B}{\|\tau_A\|_2 \cdot \|\tau_B\|_2} \in [-1, 1]$$

This metric probes the angular alignment between task-specific directions in parameter space. High similarity indicates shared representational updates; near-orthogonality suggests disentangled task pathways.

**Analyzing Task Vector Relationships via Cosine Similarity, PMI and JSD:** To analyze inter-task relationships, we work with Cosine Similarity, PMI, and JSD. While **Cosine Similarity** is a commonly used metric for comparing vector representations, it falls short in capturing nuanced differences in model behavior when applied to classification probability distributions. Cosine only measures the angular similarity between two vectors and is therefore invariant to vector magnitude. Hence, two models assigning vastly different probabilities but in the same proportional direction can still yield a high cosine score, misleadingly implying strong similarity. This limitation becomes evident in our experimental heatmap (Figure 2a), where task relationships are not clearly differentiated as many unrelated tasks appear spuriously similar due to their shared vector directionality. Moreover, cosine

similarity does not adequately account for uncertainty or confidence in model outputs.

To address these issues, we used Pointwise Mutual Information (**PMI**) and Jensen-Shannon Divergence (**JSD**), which offer better theoretical grounding and practical discriminability. As shown in Figures 2b and 2c, PMI captures directional alignment of model predictions with respect to task-specific specialization, while JSD provides a symmetric and robust comparison of output distributions. These metrics yield much more interpretable heatmaps where related tasks cluster more meaningfully and task-specific behaviors are more distinctly captured.

Concretely, the cosine heatmap appears overly uniform—masking important task groupings—whereas the PMI and JSD maps each expose clear blocks of high intra-group similarity and low inter-group coupling. These results confirm that, for fine-grained task-similarity assessment in large models, information-theoretic measures substantially outperform simple angular alignment.

Below figure 2 visualizes the effects (on SGLUE tasks), comparing cosine, PMI, and JSD heatmaps to illustrate their differing sensitivity to inter-task relationships.

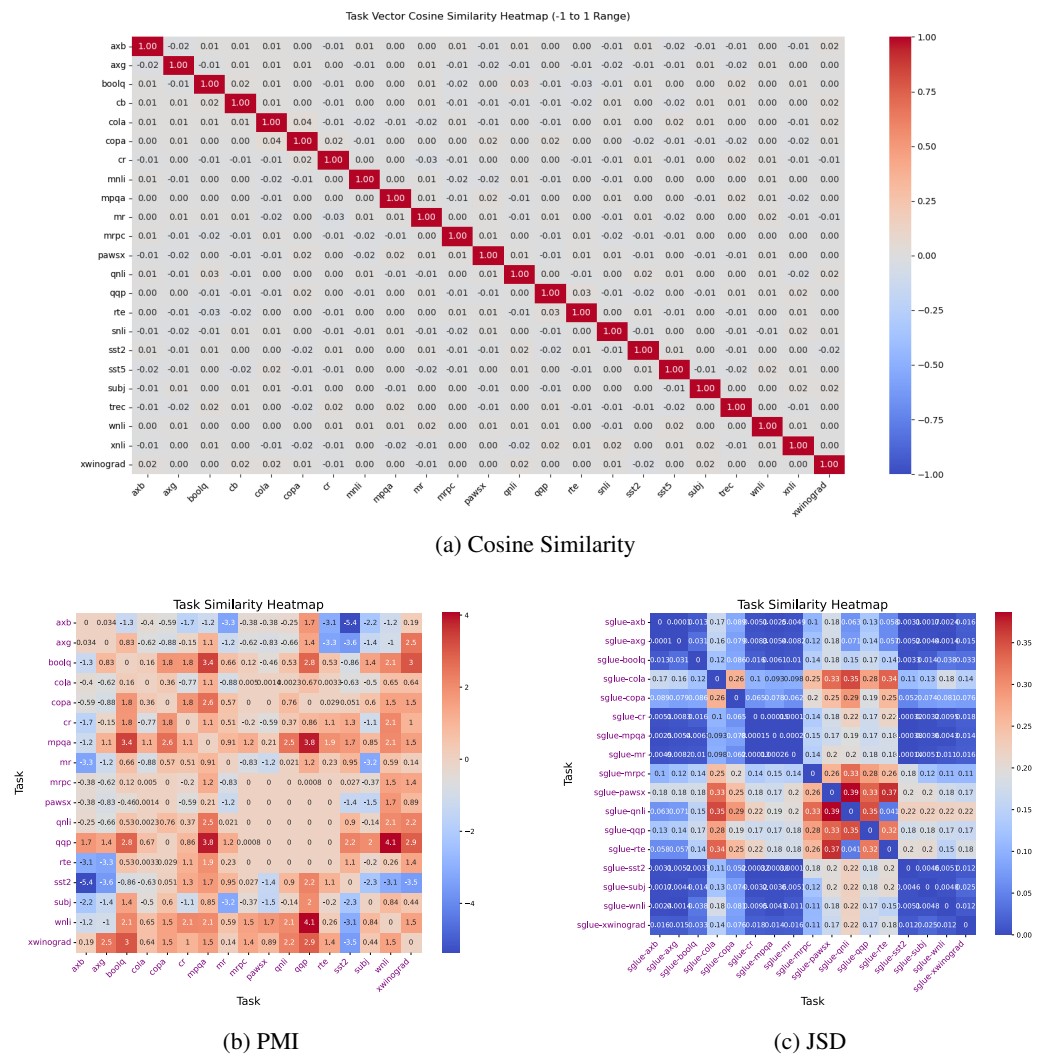

(a) Cosine Similarity

(b) PMI

(c) JSD

Figure 2: Comparison of task similarity metrics using cosine similarity (top), and PMI and JSD-based heatmaps (bottom). Cosine scores are generally low and fail to distinguish task structure. PMI highlights asymmetric task alignment. JSD offers symmetric, bounded divergence and reveals clearer task groupings across models.

## D.2 ALGORITHMS FOR COMPUTING PMI AND JSD

**Algorithm 1** performs fine-tuning by linearizing the model around its pretrained parameters. Instead of recomputing the full forward pass, it uses a Jacobian-vector product (JVP) to approximate the effect of parameter updates, allowing faster gradient-based updates in the "tangent space" of the original model.

---

**Algorithm 1: Linearized (Tangent-Space) Fine-Tuning**

**Require:** Pretrained weights $\theta_0$, dataset $D_t$

1:    Initialize $\theta \leftarrow \theta_0$
2:    **while** not converged **do**
3:      Sample mini-batch $(x, y) \sim D_t$
4:      Compute base output $o_0 = f(x; \theta_0)$
5:      Compute JVP: $g = \mathrm{JVP}\big(f(\cdot; \theta_0), \theta - \theta_0; x\big)$
6:      $\hat{o} = o_0 + g$
7:      $\theta \leftarrow \theta - \eta \nabla_\theta \ell(\hat{o}, y)$
8:    **end while**
9:    **return** $\theta_t^{\mathrm{lin}*}$

---

**Algorithm 2** quantifies how similarly two models $M_A$ and $M_B$ score the same labeled examples, using a pointwise mutual information (PMI)–inspired score. By averaging the log-ratio of predicted probabilities on each other's held-out data, it produces a symmetric similarity score $S_{AB}$.

---

**Algorithm 2: PMI-Based Inter-Model Similarity $S_{AB}$**

**Require:** Models $M_A$, $M_B$; datasets $D^A$, $D^B$
**Ensure:** Similarity score $S_{AB}$

1:    Initialize accumulator $\mathtt{sum}_B \leftarrow 0$
2:    **for all** $(x, y) \in D^B$ **do**
3:      Compute $p_A \leftarrow M_A(x)$ and extract $p_A(y)$
4:      Compute $p_B \leftarrow M_B(x)$ and extract $p_B(y)$
5:      Update $\mathtt{sum}_B \mathrel{+}= \log\left(\frac{p_A(y)}{p_B(y)}\right)$
6:    **end for**
7:    Set $\Delta_B \leftarrow \frac{1}{|D^B|} \cdot \mathtt{sum}_B$
8:    Initialize accumulator $\mathtt{sum}_A \leftarrow 0$
9:    **for all** $(x, y) \in D^A$ **do**
10:     Compute $p_A \leftarrow M_A(x)$ and extract $p_A(y)$
11:     Compute $p_B \leftarrow M_B(x)$ and extract $p_B(y)$
12:     Update $\mathtt{sum}_A \mathrel{+}= \log\left(\frac{p_B(y)}{p_A(y)}\right)$
13:    **end for**
14:   Set $\Delta_A \leftarrow \frac{1}{|D^A|} \cdot \mathtt{sum}_A$
15:   **return** $S_{AB} \leftarrow \frac{1}{2}(\Delta_A + \Delta_B)$

---

**Algorithm 3** computes the average Jensen–Shannon divergence between the predictive distributions of two models $M_A$ and $M_B$ across a shared dataset. Uses softmax outputs to measure how differently the models assign probabilities.

**Algorithm 3: Jensen–Shannon Divergence (JSD) for Model Comparison**

**Require:** Two models $M_A, M_B$; dataset $D$
**Ensure:** Average JSD value $J\bar{S}D$

1:    Initialize `total_jsd` $\leftarrow 0$
2:    **for** each input $(x, y) \in D$ **do**
3:       $P \leftarrow \mathrm{softmax}(M_A(x))$                   {Predictive distribution from $M_A$}
4:       $Q \leftarrow \mathrm{softmax}(M_B(x))$                   {Predictive distribution from $M_B$}
5:       $M \leftarrow \frac{1}{2}(P + Q)$                         {Mixture distribution}
6:       $KL_P \leftarrow \sum_i P_i \log\left(\frac{P_i}{M_i}\right)$
7:       $KL_Q \leftarrow \sum_i Q_i \log\left(\frac{Q_i}{M_i}\right)$
8:       $JSD(x) \leftarrow \frac{1}{2}(KL_P + KL_Q)$
9:       `total_jsd` $\leftarrow$ `total_jsd` $+ JSD(x)$
10:   **end for**
11:   **return** $J\bar{S}D \leftarrow \frac{\texttt{total\_jsd}}{|D|}$

# E    EXPERIMENTAL DETAILS

All the experiments are conducted in a standardized and uniform environment to ensure reproducibility and cost-effectiveness. We finetune the models for one epoch on each dataset split, leveraging 8 NVIDIA H100 GPUs in bf16 precision. We use a per-device train batch size of 1, and using AdamW optimizer with a learning rate of 2 x $10^{-5}$, weight decay 0.01, and gradient accumulation of 1. A linear learning-rate decay schedule is applied with a linear warmup over the first 3 % of total steps. To maximize memory efficiency, we enable gradient checkpointing and used DDP. The workloads are largely of 3 types, specifications and details of each are listed below.

**Fine-Tuning on Task Pool Datasets:** The objective of the approach is to find a final mixture from a large set of datasets which target different tasks. The pre-trained causal language model was used as the base model that was fine-tuned on each individual task. This stage follows the same configuration, with the following modification: models are finetuned for 3 epochs using an effective batch size of 64 and a cosine learning rate decay. A higher weight decay of 0.1 was applied, and all 8 GPUs were utilized in a Data Parallel setting. The goal is to train individual models on 316 distinct task drawn from diverse target sub-mixtures (T0, Flan2021, CoT, TULU, SGlue). All fine-tunings are full-parameter with no freezing or adapters.

**Similarity Matrix Computation :** We propose the use of two primary metrics, namely, 1) PMI and 2) JSD, although we arrive at the same by exhaustive experiments and analysis of other similarity measures and conclude with the efficacy of the two metrics. The **PMI** matrix computation, as illustrated in Algorithm 2 in Section D.2, is implemented similarly with optimizations at the PyTorch GPU and CPU multiprocessing level to speed up the computation of pairwise similarity scores due to the higher number of inferences required. We acquired the **JSD** matrix following the procedure outlined in Algorithm 3 in Section D.2. To optimize computation, we first precompute and store each model's self-distribution ($P_{X \to X}$) and cross-distribution ($P_{X \to Y}$) across all tasks to prevent redundant forward passes. Distribution computation is vectorized by batching samples per task into single forward passes and all pairwise JSD values were calculated in parallel. A total of $\frac{n(n-1)}{2}$ pairs were computed in both the cases, due to the inherent symmetric nature of the metric matrices, where $n$ is the number of tasks.

**Fine-Tuning on Final Mixture :** This phase follows the same environment and base hyperparameters configuration described earlier, with modifications tailored to the final mixture evaluation. The mixture dataset acquired from the set of tasks using our proposed solution has to be evaluated against recognized benchmarks, for which the mixture dataset is used to fine-tune a Llama-2-7B model for a single epoch with an effective batch size of 8, a learning rate of 2 x $10^{-5}$ and gradient accumulation at every 8th step. A weight decay of 0.01 was used along with cosine learning rate decay and all 8 GPUs were utilized in a Data Parallel setting. Same hyperparameters and environment configuration was used when fine-tuning on Mistral-7B to showcase the relevance of the base model in the experimental results from our proposed mixture. We further explore mixture scale by evaluating training on subsets of varying sizes (25K, 50K and 100K) and examine performance sensitivity to batch size by comparing runs with effective batch size of 8. Additionally, for the 25K and 50K subsets, we conducted experiments with different values of $\beta$ and $\lambda$ to analyze their influence on mixture composition in both PMI-based and JSD-based submix selection strategies.

# F  ADDITIONAL RESULTS

Table 3: Llama-2-7b: Instruction-tuning performance on MMLU and Leaderboard subsets with $\beta = 20, \lambda = 10$ using batch size 8.

| Dataset | | MMLU | Leaderboard | | | | | |
|---|---|---|---|---|---|---|---|---|
| Size | Method | | BBH | GPQA | IFEval | Math | MMLU-Pro | MUSR |
| **25K** | | | | | | | | |
| 25K | Random | $0.3913_{\pm0.0040}$ | $0.3482_{\pm0.0059}$ | $0.2626_{\pm0.0128}$ | $\mathbf{0.3729}_{\pm N/A}$ | $0.0098_{\pm0.0027}$ | $\mathbf{0.1877}_{\pm0.0036}$ | $0.3677_{\pm0.0172}$ |
| 25K | Uniform | $0.3479_{\pm0.0039}$ | $0.3501_{\pm0.0059}$ | $0.2701_{\pm0.0129}$ | $0.3501_{\pm N/A}$ | $0.0151_{\pm0.0034}$ | $0.1768_{\pm0.0035}$ | $\mathbf{0.4127}_{\pm0.0175}$ |
| 25K | EPM | $0.3802_{\pm0.0040}$ | $0.3593_{\pm0.0059}$ | $0.2601_{\pm0.0127}$ | $0.3405_{\pm N/A}$ | $0.0151_{\pm0.0033}$ | $0.1836_{\pm0.0035}$ | $0.4286_{\pm0.0177}$ |
| 25K | Ours (PMI) | $\mathbf{0.4242}_{\pm0.0040}$ | $\mathbf{0.3598}_{\pm0.0059}$ | $0.2718_{\pm0.0129}$ | $0.3561_{\pm N/A}$ | $0.0136_{\pm0.0032}$ | $\mathbf{0.1877}_{\pm0.0036}$ | $0.4008_{\pm0.0174}$ |
| 25K | Ours (JSD) | $0.3926_{\pm0.0040}$ | $0.3454_{\pm0.0059}$ | $\mathbf{0.2785}_{\pm0.0130}$ | $0.3465_{\pm N/A}$ | $0.0151_{\pm0.0034}$ | $0.1790_{\pm0.0035}$ | $0.4021_{\pm0.0175}$ |
| **50K** | | | | | | | | |
| 50K | Random | $0.4108_{\pm0.0040}$ | $0.3565_{\pm0.0060}$ | $0.2668_{\pm0.0128}$ | $0.3681_{\pm N/A}$ | $0.0144_{\pm0.0033}$ | $0.1881_{\pm0.0036}$ | $0.3770_{\pm0.0172}$ |
| 50K | Uniform | $0.3725_{\pm0.0040}$ | $0.3480_{\pm0.0059}$ | $0.2785_{\pm0.0130}$ | $\mathbf{0.4041}_{\pm N/A}$ | $0.0181_{\pm0.0037}$ | $0.1896_{\pm0.0036}$ | $\mathbf{0.4206}_{\pm0.0176}$ |
| 50K | EPM | $0.3801_{\pm0.0040}$ | $0.3532_{\pm0.0059}$ | $0.2634_{\pm0.0128}$ | $0.3507_{\pm N/A}$ | $0.0128_{\pm0.0031}$ | $0.1799_{\pm0.0035}$ | $0.4206_{\pm0.0176}$ |
| 50K | Ours (PMI) | $\mathbf{0.4156}_{\pm0.0040}$ | $0.3619_{\pm0.0060}$ | $0.2794_{\pm0.0130}$ | $0.3417_{\pm N/A}$ | $\mathbf{0.0189}_{\pm0.0037}$ | $0.1856_{\pm0.0035}$ | $0.3876_{\pm0.0174}$ |
| 50K | Ours (JSD) | $0.4074_{\pm0.0040}$ | $\mathbf{0.3624}_{\pm0.0060}$ | $\mathbf{0.2802}_{\pm0.0130}$ | $0.3525_{\pm N/A}$ | $0.0098_{\pm0.0027}$ | $\mathbf{0.1927}_{\pm0.0036}$ | $\mathbf{0.4206}_{\pm0.0176}$ |
| **100K** | | | | | | | | |
| 100K | Random | $0.3816_{\pm0.0040}$ | $0.3458_{\pm0.0059}$ | $0.2621_{\pm0.0128}$ | $0.3705_{\pm N/A}$ | $0.0113_{\pm0.0029}$ | $\mathbf{0.1893}_{\pm0.0036}$ | $0.4101_{\pm0.0176}$ |
| 100K | Uniform | $0.3953_{\pm0.0040}$ | $0.3569_{\pm0.0060}$ | $0.2710_{\pm0.0129}$ | $\mathbf{0.3801}_{\pm N/A}$ | $0.0189_{\pm0.0037}$ | $0.1890_{\pm0.0036}$ | $0.3730_{\pm0.0172}$ |
| 100K | EPM | $0.3915_{\pm0.0040}$ | $0.3439_{\pm0.0059}$ | $0.2844_{\pm0.0131}$ | $0.3717_{\pm N/A}$ | $0.0098_{\pm0.0027}$ | $0.1873_{\pm0.0036}$ | $0.4259_{\pm0.0176}$ |
| 100K | Ours (PMI) | $0.4021_{\pm0.0040}$ | $\mathbf{0.3633}_{\pm0.0059}$ | $0.2626_{\pm0.0127}$ | $0.3525_{\pm N/A}$ | $0.0166_{\pm0.0035}$ | $0.1894_{\pm0.0035}$ | $0.3902_{\pm0.0174}$ |
| 100K | Ours (JSD) | $\mathbf{0.4256}_{\pm0.0040}$ | $0.3598_{\pm0.0060}$ | $\mathbf{0.2894}_{\pm0.0131}$ | $0.3769_{\pm N/A}$ | $\mathbf{0.0273}_{\pm0.0035}$ | $0.1923_{\pm0.0036}$ | $0.4101_{\pm0.0176}$ |

Table 4: Mistral-7B: Instruction-tuning performance on MMLU and Leaderboard subsets with $\beta = 20, \lambda = 10$ using batch size 8.

| Dataset | | MMLU | Leaderboard | | | | | |
|---|---|---|---|---|---|---|---|---|
| Size | Method | | BBH | GPQA | IFEval | Math | MMLU-Pro | MUSR |
| **25K** | | | | | | | | |
| 25K | Random | $\mathbf{0.4539}_{\pm0.0041}$ | $\mathbf{0.3701}_{\pm0.0060}$ | $0.2760_{\pm0.0130}$ | $0.4197_{\pm N/A}$ | $0.0120_{\pm0.0031}$ | $\mathbf{0.1762}_{\pm0.0035}$ | $0.4101_{\pm0.0175}$ |
| 25K | Uniform | $0.4376_{\pm0.0041}$ | $0.3628_{\pm0.0060}$ | $0.2601_{\pm0.0127}$ | $0.4029_{\pm N/A}$ | $\mathbf{0.0159}_{\pm0.0034}$ | $0.1735_{\pm0.0035}$ | $\mathbf{0.4458}_{\pm0.0178}$ |
| 25K | EPM | $0.4364_{\pm0.0041}$ | $0.3355_{\pm0.0060}$ | $0.2869_{\pm0.0131}$ | $\mathbf{0.4281}_{\pm N/A}$ | $0.0121_{\pm0.0030}$ | $0.1498_{\pm0.0033}$ | $0.3492_{\pm0.0168}$ |
| 25K | Ours (PMI) | $0.3903_{\pm0.0040}$ | $0.3244_{\pm0.0058}$ | $0.2626_{\pm0.0128}$ | $0.3297_{\pm N/A}$ | $0.0128_{\pm0.0031}$ | $0.1503_{\pm0.0033}$ | $0.3836_{\pm0.0174}$ |
| 25K | Ours (JSD) | $0.3783_{\pm0.0040}$ | $0.3420_{\pm0.0060}$ | $\mathbf{0.2878}_{\pm0.0131}$ | $0.3525_{\pm N/A}$ | $0.0129_{\pm0.0031}$ | $0.1742_{\pm0.0034}$ | $0.3929_{\pm0.0174}$ |
| **50K** | | | | | | | | |
| 50K | Random | $0.4177_{\pm0.0040}$ | $0.3446_{\pm0.0059}$ | $0.2659_{\pm0.0128}$ | $0.4113_{\pm N/A}$ | $0.0106_{\pm0.0028}$ | $0.1733_{\pm0.0035}$ | $0.3836_{\pm0.0175}$ |
| 50K | Uniform | $\mathbf{0.4452}_{\pm0.0041}$ | $0.3479_{\pm0.0059}$ | $0.2651_{\pm0.0128}$ | $0.4161_{\pm N/A}$ | $0.0151_{\pm0.0033}$ | $0.1799_{\pm0.0035}$ | $0.3823_{\pm0.0172}$ |
| 50K | EPM | $0.4405_{\pm0.0041}$ | $0.3413_{\pm0.0059}$ | $0.2701_{\pm0.0129}$ | $\mathbf{0.4293}_{\pm N/A}$ | $0.0174_{\pm0.0036}$ | $0.1871_{\pm0.0036}$ | $0.4034_{\pm0.0174}$ |
| 50K | Ours (PMI) | $0.4228_{\pm0.0040}$ | $0.3492_{\pm0.0058}$ | $\mathbf{0.2735}_{\pm0.0129}$ | $0.3094_{\pm N/A}$ | $\mathbf{0.0174}_{\pm0.0036}$ | $0.1758_{\pm0.0035}$ | $0.4259_{\pm0.0176}$ |
| 50K | Ours (JSD) | $0.4138_{\pm0.0040}$ | $\mathbf{0.3498}_{\pm0.0059}$ | $0.2567_{\pm0.0127}$ | $0.4065_{\pm N/A}$ | $0.0159_{\pm0.0034}$ | $\mathbf{0.1898}_{\pm0.0035}$ | $0.3890_{\pm0.0173}$ |
| **100K** | | | | | | | | |
| 100K | Random | $0.4476_{\pm0.0041}$ | $0.3416_{\pm0.0060}$ | $0.2542_{\pm0.0126}$ | $\mathbf{0.4388}_{\pm N/A}$ | $0.0186_{\pm0.0038}$ | $0.1730_{\pm0.0034}$ | $0.4048_{\pm0.0175}$ |
| 100K | Uniform | $0.4486_{\pm0.0041}$ | $0.3532_{\pm0.0059}$ | $\mathbf{0.2661}_{\pm0.0128}$ | $0.3741_{\pm N/A}$ | $0.0174_{\pm0.0036}$ | $0.1724_{\pm0.0034}$ | $0.3810_{\pm0.0173}$ |
| 100K | EPM | $0.4505_{\pm0.0041}$ | $0.3578_{\pm0.0060}$ | $0.2466_{\pm0.0125}$ | $\mathbf{0.4388}_{\pm N/A}$ | $0.0174_{\pm0.0036}$ | $0.1859_{\pm0.0035}$ | $0.4074_{\pm0.0175}$ |
| 100K | Ours (PMI) | $\mathbf{0.5476}_{\pm0.0040}$ | $0.3388_{\pm0.0058}$ | $0.2508_{\pm0.0126}$ | $0.3369_{\pm N/A}$ | $0.0136_{\pm0.0032}$ | $0.1810_{\pm0.0035}$ | $0.4081_{\pm0.0176}$ |
| 100K | Ours (JSD) | $0.5301_{\pm0.0040}$ | $\mathbf{0.3591}_{\pm0.0060}$ | $0.2667_{\pm0.0127}$ | $0.4137_{\pm N/A}$ | $\mathbf{0.0189}_{\pm0.0037}$ | $\mathbf{0.1953}_{\pm0.0035}$ | $\mathbf{0.4140}_{\pm0.0175}$ |

| | | | |
|---|---|---|---|
| highest accuracy | 2nd highest accuracy | 3rd highest accuracy. |

We observe that for **LLaMA** as the base model, increasing the number of instances in the mixture has negligible impact on performance when using $\beta = 20$ and $\lambda = 10$. However, in the case of **Mistral**, the same configuration leads to a substantial improvement, where our **PMI-based method** yields **at least 10% higher accuracy on MMLU** compared to heuristic-driven methods. This strongly indicates that **PMI scales more effectively** with larger mixtures, leveraging the increased data volume to improve instruction tuning performance.

On the other hand, methods that rely on **heuristics** tend to perform better with **smaller instance sizes**. The reduced size helps **control the randomness** in mixture construction, suggesting that such heuristic approaches **do not scale well** as the number of instances increases. This confirms that their design may lack robustness in high-complexity or large-scale scenarios, where principled methods like PMI show a clear advantage.

Table 5: Llama-2-7b: Instruction-tuning performance on MMLU and Leaderboard subsets with varying $\beta$=20, $\lambda$=10 using batch size 64.

| Dataset | | MMLU | Leaderboard | | | | | |
|---|---|---|---|---|---|---|---|---|
| Size | Method | | BBH | GPQA | IFEval | Math | MMLU-Pro | MUSR |
| **25K** | | | | | | | | |
| 25K | Random | $\mathbf{0.4004}_{\pm 0.0040}$ | $\mathbf{0.3602}_{\pm 0.0059}$ | $\mathbf{0.2928}_{\pm 0.0132}$ | $0.3357_{\pm N/A}$ | $\mathbf{0.0166}_{\pm 0.0035}$ | $\mathbf{0.1924}_{\pm 0.0036}$ | $0.3889_{\pm 0.0173}$ |
| 25K | Uniform | $0.3987_{\pm 0.0040}$ | $0.3525_{\pm 0.0059}$ | $0.2710_{\pm 0.0129}$ | $0.3441_{\pm N/A}$ | $0.0159_{\pm 0.0034}$ | $0.1832_{\pm 0.0035}$ | $\mathbf{0.4220}_{\pm 0.0176}$ |
| 25K | EPM | $0.3970_{\pm 0.0040}$ | $0.3468_{\pm 0.0059}$ | $0.2685_{\pm 0.0128}$ | $\mathbf{0.3681}_{\pm N/A}$ | $0.0128_{\pm 0.0031}$ | $0.1853_{\pm 0.0035}$ | $0.4140_{\pm 0.0176}$ |
| 25K | Ours (PMI) | $0.3917_{\pm 0.0040}$ | $0.3479_{\pm 0.0059}$ | $0.2676_{\pm 0.0128}$ | $0.3477_{\pm N/A}$ | $0.0106_{\pm 0.0028}$ | $0.1918_{\pm 0.0036}$ | $0.3889_{\pm 0.0174}$ |
| 25K | Ours (JSD) | $0.4057_{\pm 0.0040}$ | $0.3517_{\pm 0.0059}$ | $0.2886_{\pm 0.0131}$ | $0.3273_{\pm N/A}$ | $0.0121_{\pm 0.0030}$ | $0.1849_{\pm 0.0035}$ | $0.3995_{\pm 0.0175}$ |
| **50K** | | | | | | | | |
| 50K | Random | $0.3761_{\pm 0.0040}$ | $0.3515_{\pm 0.0059}$ | $0.2810_{\pm 0.0130}$ | $0.3549_{\pm N/A}$ | $0.0113_{\pm 0.0029}$ | $0.1845_{\pm 0.0035}$ | $0.3796_{\pm 0.0172}$ |
| 50K | Uniform | $0.3923_{\pm 0.0040}$ | $\mathbf{0.3612}_{\pm 0.0059}$ | $0.2710_{\pm 0.0129}$ | $0.3693_{\pm N/A}$ | $0.0121_{\pm 0.0030}$ | $0.1875_{\pm 0.0036}$ | $0.4206_{\pm 0.0177}$ |
| 50K | EPM | $\mathbf{0.4029}_{\pm 0.0040}$ | $0.3461_{\pm 0.0059}$ | $0.2710_{\pm 0.0129}$ | $\mathbf{0.3885}_{\pm N/A}$ | $0.0151_{\pm 0.0034}$ | $0.1869_{\pm 0.0036}$ | $0.4325_{\pm 0.0177}$ |
| 50K | Ours (PMI) | $0.3748_{\pm 0.0040}$ | $0.3562_{\pm 0.0059}$ | $\mathbf{0.2878}_{\pm 0.0131}$ | $0.3441_{\pm N/A}$ | $\mathbf{0.0159}_{\pm 0.0034}$ | $0.1896_{\pm 0.0036}$ | $0.3929_{\pm 0.0174}$ |
| 50K | Ours (JSD) | $0.3758_{\pm 0.0040}$ | $0.3543_{\pm 0.0060}$ | $0.2676_{\pm 0.0128}$ | $0.3741_{\pm N/A}$ | $0.0136_{\pm 0.0032}$ | $\mathbf{0.1902}_{\pm 0.0036}$ | $\mathbf{0.4220}_{\pm 0.0176}$ |
| **100K** | | | | | | | | |
| 100K | Random | $0.3816_{\pm 0.0040}$ | $0.3458_{\pm 0.0059}$ | $0.2651_{\pm 0.0128}$ | $0.3705_{\pm N/A}$ | $0.0113_{\pm 0.0029}$ | $0.1893_{\pm 0.0036}$ | $0.4101_{\pm 0.0176}$ |
| 100K | Uniform | $0.3953_{\pm 0.0040}$ | $0.3569_{\pm 0.0060}$ | $0.2710_{\pm 0.0129}$ | $\mathbf{0.3801}_{\pm N/A}$ | $\mathbf{0.0189}_{\pm 0.0037}$ | $0.1890_{\pm 0.0036}$ | $0.3730_{\pm 0.0172}$ |
| 100K | EPM | $0.3915_{\pm 0.0040}$ | $0.3439_{\pm 0.0059}$ | $\mathbf{0.2844}_{\pm 0.0131}$ | $0.3717_{\pm N/A}$ | $0.0098_{\pm 0.0027}$ | $0.1873_{\pm 0.0036}$ | $\mathbf{0.4259}_{\pm 0.0176}$ |
| 100K | Ours (PMI) | $0.4017_{\pm 0.0040}$ | $0.3591_{\pm 0.0060}$ | $0.2827_{\pm 0.0131}$ | $0.3213_{\pm N/A}$ | $0.0166_{\pm 0.0035}$ | $0.1854_{\pm 0.0035}$ | $0.4021_{\pm 0.0175}$ |
| 100K | Ours (JSD) | $\mathbf{0.4165}_{\pm 0.0040}$ | $\mathbf{0.3609}_{\pm 0.0060}$ | $0.2827_{\pm 0.0131}$ | $0.3585_{\pm N/A}$ | $0.0151_{\pm 0.0034}$ | $\mathbf{0.1893}_{\pm 0.0036}$ | $0.3981_{\pm 0.0176}$ |

Table 6: Mistral-7B: Instruction-tuning performance on MMLU and Leaderboard subsets with varying $\beta = 20, \lambda = 10$ using batch size 64.

| Dataset | | MMLU | Leaderboard | | | | | |
|---|---|---|---|---|---|---|---|---|
| Size | Method | | BBH | GPQA | IFEval | Math | MMLU-Pro | MUSR |
| **25K** | | | | | | | | |
| 25K | Random | $0.5541_{\pm 0.0040}$ | $\mathbf{0.4227}_{\pm 0.0061}$ | $0.2903_{\pm 0.0132}$ | $0.4544_{\pm N/A}$ | $0.0227_{\pm 0.0041}$ | $\mathbf{0.2578}_{\pm 0.0040}$ | $\mathbf{0.4484}_{\pm 0.0179}$ |
| 25K | Uniform | $\mathbf{0.5600}_{\pm 0.0040}$ | $0.4055_{\pm 0.0061}$ | $0.2685_{\pm 0.0128}$ | $\mathbf{0.4592}_{\pm N/A}$ | $0.0242_{\pm 0.0042}$ | $0.2557_{\pm 0.0040}$ | $0.4259_{\pm 0.0176}$ |
| 25K | EPM | $0.5449_{\pm 0.0040}$ | $0.4152_{\pm 0.0062}$ | $0.2735_{\pm 0.0129}$ | $0.4376_{\pm N/A}$ | $0.0219_{\pm 0.0040}$ | $0.2345_{\pm 0.0039}$ | $0.4418_{\pm 0.0178}$ |
| 25K | Ours (PMI) | $0.5383_{\pm 0.0040}$ | $0.4171_{\pm 0.0062}$ | $0.2626_{\pm 0.0128}$ | $0.3921_{\pm N/A}$ | $0.0211_{\pm 0.0039}$ | $0.2485_{\pm 0.0039}$ | $0.3876_{\pm 0.0175}$ |
| 25K | Ours (JSD) | $0.5400_{\pm 0.0040}$ | $0.4180_{\pm 0.0062}$ | $\mathbf{0.2928}_{\pm 0.0132}$ | $0.4544_{\pm N/A}$ | $\mathbf{0.0264}_{\pm 0.0044}$ | $0.2462_{\pm 0.0039}$ | $0.4180_{\pm 0.0178}$ |
| **50K** | | | | | | | | |
| 50K | Random | $0.5524_{\pm 0.0040}$ | $0.4044_{\pm 0.0061}$ | $\mathbf{0.2878}_{\pm 0.0131}$ | $0.4844_{\pm N/A}$ | $\mathbf{0.0272}_{\pm 0.0045}$ | $0.2620_{\pm 0.0040}$ | $0.3995_{\pm 0.0174}$ |
| 50K | Uniform | $\mathbf{0.5585}_{\pm 0.0040}$ | $0.4062_{\pm 0.0061}$ | $0.2727_{\pm 0.0129}$ | $\mathbf{0.4940}_{\pm N/A}$ | $0.0257_{\pm 0.0043}$ | $\mathbf{0.2702}_{\pm 0.0040}$ | $0.4272_{\pm 0.0178}$ |
| 50K | EPM | $0.5541_{\pm 0.0040}$ | $0.4294_{\pm 0.0062}$ | $0.2861_{\pm 0.0131}$ | $0.4676_{\pm N/A}$ | $0.0189_{\pm 0.0037}$ | $0.2612_{\pm 0.0040}$ | $\mathbf{0.4458}_{\pm 0.0179}$ |
| 50K | Ours (PMI) | $0.5499_{\pm 0.0040}$ | $\mathbf{0.4135}_{\pm 0.0061}$ | $0.2861_{\pm 0.0131}$ | $0.3825_{\pm N/A}$ | $0.0181_{\pm 0.0037}$ | $0.2479_{\pm 0.0039}$ | $0.4378_{\pm 0.0178}$ |
| 50K | Ours (JSD) | $0.5389_{\pm 0.0040}$ | $0.3543_{\pm 0.0060}$ | $0.2676_{\pm 0.0128}$ | $0.3741_{\pm N/A}$ | $0.0136_{\pm 0.0032}$ | $0.1902_{\pm 0.0036}$ | $0.4220_{\pm 0.0176}$ |
| **100K** | | | | | | | | |
| 100K | Random | $0.4476_{\pm 0.0041}$ | $0.3416_{\pm 0.0060}$ | $0.2542_{\pm 0.0126}$ | $\mathbf{0.4388}_{\pm N/A}$ | $0.0196_{\pm 0.0038}$ | $0.1730_{\pm 0.0034}$ | $0.4048_{\pm 0.0175}$ |
| 100K | Uniform | $0.4486_{\pm 0.0041}$ | $0.3532_{\pm 0.0059}$ | $0.2668_{\pm 0.0128}$ | $0.3741_{\pm N/A}$ | $0.0174_{\pm 0.0036}$ | $0.1724_{\pm 0.0034}$ | $0.3810_{\pm 0.0173}$ |
| 100K | EPM | $0.4505_{\pm 0.0040}$ | $0.3578_{\pm 0.0060}$ | $0.2466_{\pm 0.0125}$ | $\mathbf{0.4388}_{\pm N/A}$ | $0.0174_{\pm 0.0036}$ | $0.1859_{\pm 0.0035}$ | $0.4074_{\pm 0.0176}$ |
| 100K | Ours (PMI) | $\mathbf{0.5476}_{\pm 0.0040}$ | $\mathbf{0.4161}_{\pm 0.0062}$ | $0.2701_{\pm 0.0129}$ | $0.3501_{\pm N/A}$ | $\mathbf{0.0234}_{\pm 0.0041}$ | $\mathbf{0.2558}_{\pm 0.0040}$ | $0.4101_{\pm 0.0176}$ |
| 100K | Ours (JSD) | $0.5301_{\pm 0.0040}$ | $0.3784_{\pm 0.0059}$ | $\mathbf{0.2768}_{\pm 0.0130}$ | $0.4257_{\pm N/A}$ | $0.0204_{\pm 0.0039}$ | $0.2342_{\pm 0.0039}$ | $\mathbf{0.4206}_{\pm 0.0176}$ |

We demonstrate that a **small adjustment in batch size**-specifically increasing it to **64**-in conjunction with the use of **Mistral**, allows us to achieve performance that is **comparable to a 100K instance mixture** trained with **BS=8**, while using only a **25K instance mixture**. This setup delivers a **7% boost in performance on BBH and MMLU-Pro**, thereby validating the **efficacy of our mixture strategy**. These results suggest that, when provided with the right computational environment, our mixture formulation has the **potential to match or surpass** much larger-scale setups on major benchmarks. Furthermore, our **JSD-based mixture** shows a remarkable **13% improvement over its LLaMA variant** when deployed with Mistral and BS=64. This emphasizes the importance of **careful hyperparameter tuning** in fully realizing the benefits of the proposed mixtures.

We also observe a **consistent gain of 5–13%** across several **leaderboard benchmarks**, including **BBH, IFEval, and Math**, when the instance size is scaled from **25K to 50K** using Mistral. However, the same scaling yields only a modest **1–2% improvement with LLaMA**. Notably, increasing the instance size to **100K results in negligible performance gains** across most benchmarks for both Mistral and LLaMA, suggesting a possible **diminishing return** beyond a certain mixture size threshold.

Table 7: Llama-2-7b: Instruction-tuning performance on MMLU and Leaderboard subsets with varying $\beta$ and $\lambda$ using batch size 8.

| Dataset Size(Method) | MMLU | Leaderboard | | | | | |
|---|---|---|---|---|---|---|---|
| | | BBH | GPQA | IFEval | Math | MMLU-Pro | MUSR |
| **25K (PMI)** | | | | | | | |
| $\beta$=14954 ; $\lambda$=263 | $0.4098_{\pm0.0040}$ | $0.3637_{\pm0.0059}$ | $0.2685_{\pm0.0128}$ | $0.3405_{\pm N/A}$ | $0.0159_{\pm0.0034}$ | $0.1869_{\pm0.0036}$ | $0.3849_{\pm0.0173}$ |
| $\beta$=5273 ; $\lambda$=195 | $0.4045_{\pm0.0040}$ | $0.3536_{\pm0.0059}$ | $0.2718_{\pm0.0129}$ | $0.3609_{\pm N/A}$ | $\mathbf{0.0166}_{\pm0.0035}$ | $0.1823_{\pm0.0035}$ | $0.4021_{\pm0.0174}$ |
| $\beta$=2535 ; $\lambda$=196 | $\mathbf{0.4258}_{\pm0.0040}$ | $\mathbf{0.3659}_{\pm0.0059}$ | $0.2701_{\pm0.0129}$ | $0.3357_{\pm N/A}$ | $0.0128_{\pm0.0031}$ | $\mathbf{0.1890}_{\pm0.0036}$ | $0.3929_{\pm0.0173}$ |
| $\beta$=307 ; $\lambda$=60 | $0.3977_{\pm0.0040}$ | $0.3605_{\pm0.0059}$ | $\mathbf{0.2735}_{\pm0.0129}$ | $\mathbf{0.3681}_{\pm N/A}$ | $0.0159_{\pm0.0034}$ | $0.1881_{\pm0.0036}$ | $0.4074_{\pm0.0174}$ |
| $\beta$=19 ; $\lambda$=5 | $0.3827_{\pm0.0040}$ | $0.3576_{\pm0.0059}$ | $0.2693_{\pm0.0129}$ | $0.3381_{\pm N/A}$ | $0.0121_{\pm0.0030}$ | $0.1872_{\pm0.0036}$ | $\mathbf{0.4246}_{\pm0.0177}$ |
| **25K (JSD)** | | | | | | | |
| $\beta$=14954 ; $\lambda$=263 | $0.3929_{\pm0.0040}$ | $0.3486_{\pm0.0059}$ | $0.2626_{\pm0.0128}$ | $0.3357_{\pm N/A}$ | $\mathbf{0.0189}_{\pm0.0037}$ | $0.1828_{\pm0.0035}$ | $0.3995_{\pm0.0176}$ |
| $\beta$=5273 ; $\lambda$=195 | $0.3793_{\pm0.0040}$ | $\mathbf{0.3614}_{\pm0.0059}$ | $0.2693_{\pm0.0129}$ | $\mathbf{0.3657}_{\pm N/A}$ | $0.0166_{\pm0.0035}$ | $0.1769_{\pm0.0035}$ | $0.3981_{\pm0.0174}$ |
| $\beta$=2535 ; $\lambda$=196 | $0.3978_{\pm0.0040}$ | $0.3574_{\pm0.0059}$ | $\mathbf{0.2794}_{\pm0.0130}$ | $0.3573_{\pm N/A}$ | $0.0113_{\pm0.0029}$ | $0.1844_{\pm0.0035}$ | $\mathbf{0.4048}_{\pm0.0175}$ |
| $\beta$=307 ; $\lambda$=60 | $\mathbf{0.4188}_{\pm0.0040}$ | $0.3522_{\pm0.0060}$ | $\mathbf{0.2794}_{\pm0.0130}$ | $0.3453_{\pm N/A}$ | $0.0144_{\pm0.0033}$ | $\mathbf{0.1913}_{\pm0.0036}$ | $0.4021_{\pm0.0176}$ |
| $\beta$=19 ; $\lambda$=5 | $0.3890_{\pm0.0040}$ | $0.3545_{\pm0.0060}$ | $0.2743_{\pm0.0129}$ | $0.3525_{\pm N/A}$ | $0.0144_{\pm0.0033}$ | $0.1823_{\pm0.0035}$ | $0.3942_{\pm0.0174}$ |
| **50K (PMI)** | | | | | | | |
| $\beta$=14954 ; $\lambda$=263 | $0.3731_{\pm0.0040}$ | $0.3527_{\pm0.0059}$ | $0.2903_{\pm0.0132}$ | $\mathbf{0.3489}_{\pm N/A}$ | $0.0166_{\pm0.0035}$ | $0.1864_{\pm0.0036}$ | $0.3968_{\pm0.0174}$ |
| $\beta$=5273 ; $\lambda$=195 | $\mathbf{0.4031}_{\pm0.0040}$ | $0.3609_{\pm0.0059}$ | $0.2836_{\pm0.0131}$ | $0.3429_{\pm N/A}$ | $0.0159_{\pm0.0034}$ | $0.1869_{\pm0.0036}$ | $0.4048_{\pm0.0174}$ |
| $\beta$=2535 ; $\lambda$=196 | $0.4164_{\pm0.0040}$ | $0.3567_{\pm0.0060}$ | $0.2735_{\pm0.0129}$ | $0.3537_{\pm N/A}$ | $0.0151_{\pm0.0034}$ | $0.1906_{\pm0.0036}$ | $0.4127_{\pm0.0174}$ |
| $\beta$=307 ; $\lambda$=60 | $0.3919_{\pm0.0040}$ | $0.3637_{\pm0.0059}$ | $0.2743_{\pm0.0129}$ | $\mathbf{0.3489}_{\pm N/A}$ | $0.0091_{\pm0.0026}$ | $0.1797_{\pm0.0035}$ | $\mathbf{0.4140}_{\pm0.0176}$ |
| $\beta$=19 ; $\lambda$=5 | $0.4004_{\pm0.0040}$ | $\mathbf{0.3690}_{\pm0.0060}$ | $\mathbf{0.2936}_{\pm0.0132}$ | $0.3393_{\pm N/A}$ | $0.0159_{\pm0.0034}$ | $\mathbf{0.1921}_{\pm0.0036}$ | $0.3862_{\pm0.0174}$ |
| **50K (JSD)** | | | | | | | |
| $\beta$=14954 ; $\lambda$=263 | $0.4212_{\pm0.0040}$ | $0.3536_{\pm0.0059}$ | $0.2659_{\pm0.0128}$ | $0.3513_{\pm N/A}$ | $0.0151_{\pm0.0034}$ | $\mathbf{0.1902}_{\pm0.0036}$ | $\mathbf{0.4233}_{\pm0.0176}$ |
| $\beta$=5273 ; $\lambda$=195 | $\mathbf{0.4219}_{\pm0.0040}$ | $0.3584_{\pm0.0060}$ | $0.2659_{\pm0.0128}$ | $0.3561_{\pm N/A}$ | $\mathbf{0.0204}_{\pm0.0039}$ | $0.1877_{\pm0.0036}$ | $0.3929_{\pm0.0175}$ |
| $\beta$=2535 ; $\lambda$=196 | $0.4205_{\pm0.0040}$ | $0.3545_{\pm0.0060}$ | $\mathbf{0.2810}_{\pm0.0130}$ | $0.3513_{\pm N/A}$ | $0.0121_{\pm0.0030}$ | $0.1895_{\pm0.0036}$ | $0.4074_{\pm0.0176}$ |
| $\beta$=307 ; $\lambda$=60 | $0.4025_{\pm0.0040}$ | $0.3600_{\pm0.0059}$ | $0.2643_{\pm0.0128}$ | $0.3585_{\pm N/A}$ | $0.0144_{\pm0.0033}$ | $0.1813_{\pm0.0035}$ | $0.3823_{\pm0.0174}$ |
| $\beta$=19 ; $\lambda$=5 | $0.4039_{\pm0.0040}$ | $\mathbf{0.3650}_{\pm0.0060}$ | $0.2668_{\pm0.0128}$ | $\mathbf{0.3561}_{\pm N/A}$ | $0.0166_{\pm0.0035}$ | $0.1846_{\pm0.0035}$ | $0.4074_{\pm0.0175}$ |

Table 8: Mistral-7b: Instruction-tuning performance on MMLU and Leaderboard subsets with varying $\beta$ and $\lambda$ using batch size 8.

| Dataset Size(Method) | MMLU | Leaderboard | | | | | |
|---|---|---|---|---|---|---|---|
| | | BBH | GPQA | IFEval | Math | MMLU-Pro | MUSR |
| **25K (PMI)** | | | | | | | |
| $\beta$=14954 ; $\lambda$=263 | $\mathbf{0.4582}_{\pm0.0041}$ | $0.3579_{\pm0.0059}$ | $0.2685_{\pm0.0128}$ | $0.3237_{\pm N/A}$ | $0.0144_{\pm0.0033}$ | $0.1893_{\pm0.0036}$ | $0.3981_{\pm0.0175}$ |
| $\beta$=5273 ; $\lambda$=195 | $0.4368_{\pm0.0040}$ | $0.3498_{\pm0.0059}$ | $0.2592_{\pm0.0127}$ | $0.3621_{\pm N/A}$ | $0.0136_{\pm0.0032}$ | $0.1863_{\pm0.0035}$ | $\mathbf{0.4696}_{\pm0.0179}$ |
| $\beta$=2535 ; $\lambda$=196 | $0.4221_{\pm0.0040}$ | $0.3579_{\pm0.0060}$ | $0.2525_{\pm0.0126}$ | $0.2842_{\pm N/A}$ | $0.0136_{\pm0.0032}$ | $\mathbf{0.1912}_{\pm0.0036}$ | $0.3677_{\pm0.0172}$ |
| $\beta$=307 ; $\lambda$=60 | $0.4568_{\pm0.0041}$ | $\mathbf{0.3612}_{\pm0.0059}$ | $0.2584_{\pm0.0127}$ | $0.3177_{\pm N/A}$ | $0.0128_{\pm0.0031}$ | $0.1877_{\pm0.0036}$ | $0.4127_{\pm0.0176}$ |
| $\beta$=19 ; $\lambda$=5 | $0.4262_{\pm0.0041}$ | $0.3532_{\pm0.0060}$ | $\mathbf{0.2693}_{\pm0.0129}$ | $\mathbf{0.3669}_{\pm N/A}$ | $\mathbf{0.0151}_{\pm0.0033}$ | $0.1841_{\pm0.0035}$ | $0.4114_{\pm0.0175}$ |
| **25K (JSD)** | | | | | | | |
| $\beta$=14954 ; $\lambda$=263 | $0.4327_{\pm0.0040}$ | $0.3909_{\pm0.0061}$ | $0.2601_{\pm0.0127}$ | $0.4077_{\pm N/A}$ | $0.0128_{\pm0.0031}$ | $0.1796_{\pm0.0035}$ | $0.4378_{\pm0.0177}$ |
| $\beta$=5273 ; $\lambda$=195 | $0.4313_{\pm0.0041}$ | $0.3637_{\pm0.0060}$ | $0.2617_{\pm0.0127}$ | $0.3453_{\pm N/A}$ | $0.0166_{\pm0.0035}$ | $0.1784_{\pm0.0035}$ | $0.4220_{\pm0.0176}$ |
| $\beta$=2535 ; $\lambda$=196 | $0.4453_{\pm0.0040}$ | $\mathbf{0.3706}_{\pm0.0060}$ | $0.2601_{\pm0.0127}$ | $\mathbf{0.3969}_{\pm N/A}$ | $0.0128_{\pm0.0031}$ | $0.1728_{\pm0.0034}$ | $0.4220_{\pm0.0176}$ |
| $\beta$=307 ; $\lambda$=60 | $\mathbf{0.4568}_{\pm0.0041}$ | $0.3604_{\pm0.0060}$ | $\mathbf{0.2810}_{\pm0.0130}$ | $0.3933_{\pm N/A}$ | $\mathbf{0.0181}_{\pm0.0037}$ | $\mathbf{0.1762}_{\pm0.0035}$ | $0.4259_{\pm0.0173}$ |
| $\beta$=19 ; $\lambda$=5 | $0.3957_{\pm0.0040}$ | $0.3581_{\pm0.0059}$ | $0.2450_{\pm0.0125}$ | $0.4137_{\pm N/A}$ | $0.0136_{\pm0.0032}$ | $0.1615_{\pm0.0034}$ | $\mathbf{0.4418}_{\pm0.0176}$ |
| **50K (PMI)** | | | | | | | |
| $\beta$=14954 ; $\lambda$=263 | $0.4325_{\pm0.0041}$ | $0.3340_{\pm0.0058}$ | $0.2668_{\pm0.0128}$ | $0.3070_{\pm N/A}$ | $0.0166_{\pm0.0035}$ | $0.1902_{\pm0.0036}$ | $0.3968_{\pm0.0176}$ |
| $\beta$=5273 ; $\lambda$=195 | $\mathbf{0.4379}_{\pm0.0041}$ | $0.3539_{\pm0.0059}$ | $\mathbf{0.2886}_{\pm0.0131}$ | $\mathbf{0.3705}_{\pm N/A}$ | $\mathbf{0.0181}_{\pm0.0037}$ | $\mathbf{0.1911}_{\pm0.0036}$ | $0.3717_{\pm0.0170}$ |
| $\beta$=2535 ; $\lambda$=196 | $0.4009_{\pm0.0040}$ | $\mathbf{0.3581}_{\pm0.0060}$ | $0.2819_{\pm0.0130}$ | $0.3501_{\pm N/A}$ | $0.0174_{\pm0.0036}$ | $0.1661_{\pm0.0034}$ | $0.4127_{\pm0.0175}$ |
| $\beta$=307 ; $\lambda$=60 | $0.4336_{\pm0.0041}$ | $0.3508_{\pm0.0059}$ | $0.2785_{\pm0.0130}$ | $0.3177_{\pm N/A}$ | $0.0136_{\pm0.0032}$ | $0.1818_{\pm0.0035}$ | $0.3810_{\pm0.0171}$ |
| $\beta$=19 ; $\lambda$=5 | $0.4134_{\pm0.0040}$ | $0.3520_{\pm0.0059}$ | $0.2601_{\pm0.0127}$ | $0.3777_{\pm N/A}$ | $0.0128_{\pm0.0031}$ | $0.1661_{\pm0.0034}$ | $\mathbf{0.4193}_{\pm0.0175}$ |
| **50K (JSD)** | | | | | | | |
| $\beta$=14954 ; $\lambda$=263 | $0.4295_{\pm0.0041}$ | $\mathbf{0.3631}_{\pm0.0060}$ | $0.2592_{\pm0.0127}$ | $0.4233_{\pm N/A}$ | $0.0196_{\pm0.0038}$ | $0.1687_{\pm0.0034}$ | $0.3929_{\pm0.0173}$ |
| $\beta$=5273 ; $\lambda$=195 | $0.4372_{\pm0.0041}$ | $0.3623_{\pm0.0060}$ | $0.2727_{\pm0.0129}$ | $0.4233_{\pm N/A}$ | $0.0159_{\pm0.0034}$ | $\mathbf{0.1768}_{\pm0.0035}$ | $0.4418_{\pm0.0177}$ |
| $\beta$=2535 ; $\lambda$=196 | $0.4350_{\pm0.0041}$ | $0.3505_{\pm0.0059}$ | $0.2617_{\pm0.0127}$ | $\mathbf{0.4424}_{\pm N/A}$ | $0.0219_{\pm0.0040}$ | $0.1669_{\pm0.0034}$ | $0.3836_{\pm0.0172}$ |
| $\beta$=307 ; $\lambda$=60 | $\mathbf{0.4400}_{\pm0.0041}$ | $0.3373_{\pm0.0059}$ | $\mathbf{0.2735}_{\pm0.0129}$ | $0.4137_{\pm N/A}$ | $\mathbf{0.0227}_{\pm0.0041}$ | $0.1750_{\pm0.0035}$ | $0.3717_{\pm0.0173}$ |
| $\beta$=19 ; $\lambda$=5 | $0.4285_{\pm0.0041}$ | $0.3444_{\pm0.0058}$ | $0.2424_{\pm0.0124}$ | $0.4257_{\pm N/A}$ | $0.0106_{\pm0.0028}$ | $0.1743_{\pm0.0035}$ | $\mathbf{0.4484}_{\pm0.0179}$ |

We observe a notable improvement in convergence for Mistral over LLaMA, reflected in a consistent **2–5% boost in benchmark performance**. This underscores Mistral's enhanced compatibility with our mixture strategies.

Among the evaluated configurations, the JSD-based mixture with $\beta = 307$ and $\lambda = 60$ **emerges as the most reliable**, frequently achieving either the best or near-best results across a diverse range of datasets and evaluation metrics.

Our analysis also reveals that **PMI and JSD excel in distinct areas**. While **JSD outperforms in leaderboard subsets**-notably on **IFEval and Math**—the **PMI method leads on MMLU tasks**, demonstrating that each method has specialized strengths.

Interestingly, we find that **leaderboard metrics benefit from larger instance mixtures**, whereas **MMLU-related tasks such as BBH and GPQA plateau or even degrade** in performance when too many instances are included. This may be due to overfitting to harder instances or increased noise from larger mixtures.

We also identify that a **balanced ratio of** $\frac{\beta}{\lambda}$, such as $\beta = 307$, $\lambda = 60$, tends to **consistently outperform** other configurations. In contrast, **higher ratios** offer **strong MMLU performance but underperform on leaderboard metrics**, while **lower ratios** result in weaker performance across BBH, GPQA, and most benchmarks, likely due to their similarity to a near-uniform distribution.

## G   CODE

We provide access to anonymous version of our code: [1]Anonymous Code

---

[1]https://anonymous.4open.science/r/task-mixtures-62D3