# OpenReview forum: "TaskMixPGM: Task Mixtures via Probabilistic Graphical Modelling for Language Model Finetuning"
_ICLR.cc/2026/Conference — ICLR 2026 Conference Desk Rejected Submission_

### Official Review · Reviewer_fkK8 · 2025-10-26

**Soundness:** 1
**Presentation:** 2
**Contribution:** 2
**Rating:** 2
**Confidence:** 3

**Summary:**

This paper introduces TaskMixPGM, a novel framework for automatically determining the optimal data mixture for fine-tuning large LLMs. The core idea is to model the set of candidate tasks as a MRF and to find the optimal mixture proportions by minimizing an energy function. This energy function is designed to balance task "representativeness" and "diversity". A key contribution is the use of behavioral divergences (JSD or PMI) to define a functional similarity metric between tasks. The authors derive a closed-form solution for the optimal mixture and provide theoretical guarantees, such as weak submodularity for budgeted task selection. Experiments on Llama-2-7B and Mistral-7B show improvements on benchmarks like MMLU over simple heuristic baselines.

**Strengths:**

**Novel and Principled Formulation**: The paper proposes a novel and theoretically-grounded framework for data mixture optimization, moving beyond common heuristics. Modeling task relationships as an MRF with an energy function that balances representativeness and diversity is an elegant approach.

**Insightful Similarity Metric**: The use of "behavioral divergences" (JSD/PMI) based on model predictions is a significant strength. This captures a deeper, functional notion of task similarity compared to superficial semantic similarity, which is more relevant to model training dynamics.

**Theoretical Guarantees**: The derivation of a closed-form solution is a practical contribution, ensuring computational efficiency.

**Weaknesses:**

1. **Deficiencies in Experimental Validation**: The empirical evidence presented is insufficient to support the paper's strong claims of superiority
   - ***Lack of Transparency and Reproducibility***: The paper fails to provide crucial implementation details. There is no analysis of the learned similarity matrix S or the final mixture weights p*. Consequently, claims of interpretability are unsubstantiated, as key insights—such as which tasks are deemed most influential or how different skill domains are absent.
   -  ***Missing Hyperparameter Analysis***: The method’s core trade-off between representativeness and diversity,  β and λ, are set to fixed values (β=20, λ=10) without any justification, ablation study.
   - ***Critically Weak Baselines***: The performance is only compared against simplistic heuristics (Random, Uniform, EPM). The paper neglects to compare against far more relevant and stronger baselines discussed in its own related work section, such as RegMix, Doremi, or instruction-tuning specific methods like LESS and SMART.
   - ***Potentially Outdated Experimental Setup***: While the chosen benchmarks are standard, the experiments rely on relatively older models (Llama-2, Mistral-7B). Validating the findings on more recent and capable models (e.g., Llama-3, Qwen2.5/3) would be necessary to demonstrate the method's relevance in the current fast-paced landscape.

2. **Unreliable Definitions and Foundational Assumptions**: The theoretical framework is built upon several questionable assumptions that may not hold in practice
   - ***Flawed Definition of 'Representativeness'***: The framework defines a task's representativeness by its cumulative similarity to all other tasks. This approach risks conflating importance with generality, potentially prioritizing generic, "common-denominator" tasks while penalizing unique and specialized ones (e.g., advanced reasoning) that are critical for expanding a model's capabilities but may be dissimilar to the majority.
   - ***Ambiguous Definition of 'Task'***: The paper fails to provide a clear, operational definition of a "task." It is unclear whether a task corresponds to an domain, entire dataset (e.g., Flan), a specific sub-task (e.g., a GLUE task), or a conceptual skill.
   - ***Oversimplified Assumption of Learning Dynamics***: The method implicitly assumes that all data within a task dataset D_i contributes uniformly and positively to learning the associated skill. This decouples the optimization from the LLM's actual learning process, ignoring the reality that data quality is heterogeneous and that some examples can be noisy or even detrimental to training.

3. **Unquantified and Potentially Prohibitive Computational Cost**: The paper claims the framework is "scalable" but fails to address its significant computational overhead. Constructing the similarity matrix requires O(n) independent fine-tuning runs and O(n^2) pairwise model evaluations. This upfront cost is neither quantified in the paper (e.g., additional GPU-hours) nor discussed in terms of its practical limits, making the scalability claim unsubstantiated.

4. **Issues with Clarity, Overclaiming, and Incomplete Literature**

   - ***Clarity and Overclaiming***: The paper's argument that an MRF "inherently" balances the desired criteria is an potential overstatement; Additionally, some theoretical sections are hard to follow, and the proofs for lemmas in Section 5 and 6 lack sufficient detail.
   - ***Writing and Presentation Flaws***: The presence of a placeholder ("X.X pp") in the contributions list, redundant phrasing in the introduction ("distinct advantages" vs. "contributions"), and a long related work section that is disconnected from the experimental comparisons.

**Questions:**

My decision could be swayed if the authors can provide convincing answers to the following critical questions:

1. **Regarding Baselines**: Can you provide a strong justification for excluding state-of-the-art baselines like SMART, RegMix, or LESS from your experiments? A direct comparison is essential to properly situate your method's performance.

2. **Regarding Hyperparameters**: How was the ratio β/λ=2 determined? Could you provide a sensitivity analysis to demonstrate how this crucial trade-off impacts the selected mixture and, consequently, the final model's performance across different evaluation benchmarks?

3. **Regarding Computational Cost and Scalability**: Given the ambiguous definition of a "task," could you provide a concrete cost analysis (e.g., in total GPU-hours) for constructing the similarity matrix in your 319-task experiment? Furthermore, how do you see the method scaling when n (the number of tasks) grows larger?

4. **Regarding Interpretability**: To substantiate your claim of interpretability, could you provide a qualitative analysis of the learned task mixture? For instance, which specific tasks received the highest weights, and what does the similarity matrix reveal about the functional relationships between different task categories (e.g., reasoning vs. coding)?

---

### Official Review · Reviewer_c4AK · 2025-10-31

**Soundness:** 2
**Presentation:** 3
**Contribution:** 2
**Rating:** 2
**Confidence:** 4

**Summary:**

This paper introduces TaskMixPGM, a method for finding the data mixture for fine-tuning a Large Language Model. The authors argue past methods their method provides a formal way to find the optimal blend of tasks. The core idea is to minimize an energy function defined on tasks and task similarities, where the task's collective similarity is maximized (generally useful), and pairwise similarities are minimized (to enforce diversity). In order to solve this problem, the authors model the task as a minimization of pairwise similarities and maximize the unary potentials.

Another design is the definition of similarity, which is defined on how other tasks perform on models fine-tuned on every single task, captured with PMI or JSD.

**Strengths:**

**Mixture optimization utilizing the task-mix structure**: This method provides a rigid and theoretically-grounded way to solve the problem of task mix. The design of the method utilizes some internal structure similarities of the tasks, which avoids the cost-prohibited two-layer optimization: optimizing the task weights, optimizing the final model performance.

**Novel similarity metric design**: the similarity metric are defined on the effect of the task, instead of lexical/semantic similarity.

**Weaknesses:**

**Similarity/centrality task assumption**:
  - One assumption of this paper is to maximize the task that is mostly similar to others (while minimizing diversity). This assumption is that any skill common to many tasks must be foundational or important. I am not sure if this assumption is valid. The common "skill" can be generic.
  - On the other hand, this depends on the quality of the task pool a lot. Imagine a case where the task pool has some good tasks $(t_0, t_1, ..., t_m)$, and some bad tasks $(t_{m+1}, t_{m+2}, ..., t_{m+n})$. If $n >> m$, this method will be highly biased by the bad task pool. This thought experiment seems to show a limitation of this method.

**Significant upfront cost**: One major limitation for using the method practically is the cost needed to build its similarity matrix. The method requires one to fine-tune one separate model for every single task. For hundreds of tasks, this is a huge one-time computational cost that could be impractical and will only get worse as more tasks are added.  At least, this cost should be clearly discussed and analyzed, providing readers an idea of tradeoff.

**Complexity in implementation**: manage a large army of finetuning training can also be error-prone. The hyperparameter setup and training details may affect the model quality, which in turn can affect the similarity calculation hence the final data mix.

**Potentially Weak Baselines**: The paper proves its method is better than simple heuristics like "Random," "Uniform," and "Size-Proportional" sampling. However, it doesn't compare against the advanced data-selection methods. This makes it hard to know ifthis method is truly state-of-the-art. The author should have compared with methods presented in the relative work section, such as RegMix. Further, a pretty relevant work (https://openreview.net/forum?id=19kqoNoc2N): Data Mixing Optimization, which also mix finetuning datasets, should have been included too.


**Potential Incomplete Experiment/Writing on Task Discovery**: At line 455, there is a small description of the experiment of "Task Discovery", intends to analyze what happens when "strong" vs. "weak" tasks are added to the mix in different orders. However, I have a hard time finding the results, leaving the section feeling incomplete.

**Other presentation/writing problems**:
  - The figures in 1(a) left and 1(b) left are too small and it is unclear what are the arguments to be made there.
  - Line 159, `subscriptsxxx` should be typos

**Questions:**

The paper defines a concept called the "task vector" ($\tau(T_i)$) with an equation, but I cannot find further usage of this term again later. What's the purpose for defining this term?

---

### Official Review · Reviewer_zJEk · 2025-11-01

**Soundness:** 2
**Presentation:** 2
**Contribution:** 2
**Rating:** 4
**Confidence:** 4

**Summary:**

This paper proposed a novel data mixture optimization algorithm by minimizing an energy function over a Markov Random Field (MRF).
The method is grounded on the energy function and MRF theory and outperform uniform and random baselines on some of the benchmarks.

**Strengths:**

1. the theoretical foundation of the energy function and MRF is well grounded.

2. empirically, the proposed method outperform uniform and random baselines on some of the benchmarks.

**Weaknesses:**

1. The computation of the Task Similarity matrix $S$ involve a retraining on each of the $k$ tasks, which introduces nontrivial computational overhead. How well the method can be scaled up with the number of tasks and model sizes?

2. The proposed algorithm can be greatly depend on the constructed Task Similarity matrix $S$. How long is the model been tuned to construct a reliable $S$?

3. Since the optimized data mixture can be stronly dependent on the model-specific task similarity, how well can the proxy-model based mixture transfer to larger-scale model or the model within different architecture family?

4. The balance between Representativeness and Diversity is controlled by the hyperparameter $\lambda$ and $\beta$. While the paper lacks an ablation for selecting $\lambda, \beta$.

5. Lack of comparison to a lot of data mixture baselines [1] [2] [3]. Can you provide comparison results to them?

6. The listed "Tasks" for experiments are very general and coarse, mostly multitasking datasets. Why do you choose those coarse datasets instead of more fine-grained, domain specialized ones? Have you done the experiments on more fine-grained datasets?


[1] DoReMi: Optimizing Data Mixtures Speeds Up Language Model Pretraining

[2] DoGE: Domain Reweighting with Generalization Estimation

[3] Aioli: A Unified Optimization Framework for Language Model Data Mixing


minor: the "X.X pp improvement" in line 102-103 is confusing.

**Questions:**

According to the attached question to each point of weaknesses.

---

### Official Review · Reviewer_t7xB · 2025-11-04

**Soundness:** 2
**Presentation:** 1
**Contribution:** 1
**Rating:** 2
**Confidence:** 5

**Summary:**

This paper studies how to construct better data mixtures for SFT on multiple tasks. They propose an objective based on a MRF and a similarity matrix over the tasks, captured via Jensen-Shannon Divergence and Pointwise Mutual Information of models trained on each task. They show that this proposed method can outperform simple mixes such as uniform and random sampling.

**Strengths:**

Quality: on both Llama2-7B and Mistral-7B, the proposed method generally outperforms simple mixes such as random and uniform sampling.

Significance: Task discovery is an important and underexplored setting.

**Weaknesses:**

Originality:
- The proposed method has limited novelty. The objective in equation (1) and fine-tuning many task specific models to compute S is nearly identical to approaches like UtiliMax (https://arxiv.org/abs/2501.11747) and Skill-It (https://arxiv.org/abs/2307.14430), besides the use of JSD/PMI.

Quality:
- Since the optimization problem is constructed only over the training domains, we do not get a complete picture of evaluation with just downstream tasks; instead we should look at how the method reduces training loss and validation loss on the domains themselves. For instance, if the set of tasks were all code and we evaluated on IFEval, then even a really good mixing algorithm may not perform great, and maybe a more arbitrary mix would do well.
- The baseline mixes in the paper do not require any additional training costs. There should also be more sophisticated mixing baselines that learn a mix, such as UtiliMax and Skill-It, and cost should be normalized across these and TaskMixPGM.
- The cost of the approach is unclear. For the experiments, do you fine-tune 319 separate models, one per task, and is it for a full epoch or less? How long is each of these fine-tuning runs versus the final mixing run?
- Should include ablations on the hyperparameters $\beta$ and $\lambda$
- I do not completely agree that we want a symmetric S. Intuitively, S_ij may want to capture the influence of domain i on domain j, which is not symmetric.

Clarity: the writing is of low quality overall. For example:
- Typos: line 102 "X.X" and line 159 "xxx"
- No algorithm box for the approach is mentioned in the body
- Section 5 and 6 look incomplete and are not well-motivated.

**Questions:**

1. Can the authors report training loss and/or validation loss on the tasks to better understand the performance of the approach?
2. Can the authors compare the approach to more sophisticated baselines that learn a mixture, rather than random/uniform sampling?
3. What is the computational cost of the approach?
4. Can you provide ablations on the hyperparameters $\beta$ and $\lambda$?

---

### Official Review · Reviewer_MLAE · 2025-11-10

**Soundness:** 3
**Presentation:** 3
**Contribution:** 3
**Rating:** 6
**Confidence:** 3

**Summary:**

TaskMixPGM optimizes LLM fine-tuning mixtures by modeling task relationships as a Markov Random Field (MRF). It minimizes an energy function using behavioral similarity metrics to derive a closed-form solution balancing task representativeness and diversity.

**Strengths:**

1. It introduces a formal, energy-based optimization for task mixing, replacing common heuristics. Its closed-form solution avoids expensive iterative searches.
2. It provides theoretical guarantees and demonstrates empirical gains over baselines on the benchmarks.

**Weaknesses:**

1. The method requires fine-tuning a distinct model per task to construct the similarity matrix, which is computationally prohibitive.
2. Consequently, the approach scales poorly to a large number of tasks. This dependency on fine-tuning also restricts its applicability to scenarios outside of this paradigm, such as pre-training.
3. The solution's reliance on matrix inversion introduces potential numerical instability, particularly when dealing with ill-conditioned similarity matrices. It remains unclear how the method addresses or mitigates this potential issue.

**Questions:**

See weaknesses.

---

### Note · Program_Chairs · 2026-01-17
**Submission Desk Rejected by Program Chairs**

The following references in this submission do not refer to real documents and/or have major errors in bibliographic information:

 J. Ye, Y. Zhang, X. Liu, Z. Wang, and H. Li. Data mixing laws: Understanding and optimizing data mixtures for fine-tuning large language models. arXiv preprint arXiv:2405.06574, 2024. URL https://arxiv.org/abs/2405.06574.
Xiaoyang Jiang, Xuezhi Wang, Xinyu Li, Shuang Wu, Xuehai Qian, and Zhiwei Luo. Regmix: A data mixture framework for robust fine-tuning of large language models. URL https://arxiv.org/abs/2405.04432.